# EFFICIENT DISCOVERY OF PARETO FRONT FOR MULTI-OBJECTIVE REINFORCEMENT LEARNING

**Ruohong Liu**
University of Oxford
Oxford, UK
`ruohong.liu@eng.ox.ac.uk`

**Yuxin Pan**
The Hong Kong University of Science and Technology
Hong Kong, China
`yuxin.pan@connect.ust.hk`

**Linjie Xu**
Queen Mary University of London
London, UK
`linjie.xu@qmul.ac.uk`

**Lei Song & Jiang Bian**
Microsoft Research Asia
Beijing, China
`{lei.song, jiang.bian}@microsoft.com`

**Pengcheng You** *
Peking University
Beijing, China
`pcyou@pku.edu.cn`

**Yize Chen**
University of Alberta
Edmonton, Canada
`yize.chen@ualberta.ca`

## ABSTRACT

Multi-objective reinforcement learning (MORL) excels at handling rapidly changing preferences in tasks that involve multiple criteria, even for unseen preferences. However, previous dominating MORL methods typically generate a fixed policy set or preference-conditioned policy through multiple training iterations exclusively for sampled preference vectors, and cannot ensure the efficient discovery of the Pareto front. Furthermore, integrating preferences into the input of policy or value functions presents scalability challenges, in particular as the dimension of the state and preference space grow, which can complicate the learning process and hinder the algorithm's performance on more complex tasks. To address these issues, we propose a two-stage Pareto front discovery algorithm called Constrained MORL (C-MORL), which serves as a seamless bridge between constrained policy optimization and MORL. Concretely, a set of policies are trained in parallel in the initialization stage, with each optimized towards its individual preference over the multiple objectives. Then, to fill the remaining vacancies in the Pareto front, the constrained optimization steps are employed to maximize one objective while constraining the other objectives to exceed a predefined threshold. Empirically, compared to recent advancements in MORL methods, our algorithm achieves more consistent and superior performances in terms of hypervolume, expected utility, and sparsity on both discrete and continuous control tasks, especially with numerous objectives (up to nine objectives in our experiments). Our code is available at `https://github.com/RuohLiuq/C-MORL`.

## 1 INTRODUCTION

In many real-world control and planning problems, multiple and sometimes even conflicting objectives are getting involved. Such situations necessitate striking a better trade-off among these decision-making goals (Roijers et al., 2013; Hayes et al., 2022). For instance, in industrial control scenarios (Salvendy, 2001; Wang et al., 2023), maximizing utility and minimizing energy consumption are of particular interest as objectives to be optimized. Since different decision makers have heterogeneous preferences over these objectives, there may exist multiple Pareto-optimal policies (Roijers et al., 2014). Classical reinforcement learning (RL) methods typically involve training individual policies exclusively to align with each preference weight vector over multiple rewards (Nagabandi et al., 2018; Gupta et al., 2018). Yet it may lead to an enormous computational burden due to the overly dependence on the model retraining and fine-tuning stages. Moreover, such policies are hard

---
*Corresponding author.

to directly generalize or transfer to newer tasks (Cobbe et al., 2019; Taiga et al., 2022). Therefore, the multi-objective reinforcement learning (MORL) paradigm has drawn significant attention by reformulating these tasks for optimizing towards multiple criterion (Xu et al., 2020; Basaklar et al., 2022; Zhu et al., 2023). MORL aims to obtain either a single policy readily adapted to different preferences (Felten et al., 2023b; 2022; Teh et al., 2017) or a set of policies (Zhao & Grover, 2024; Felten et al., 2024; Röpke et al., 2024; Kim et al., 2024) aligned with their respective preferences.

One prevalent category of MORL approaches is to train a single preference-conditioned policy (Yang et al., 2019; Basaklar et al., 2022). They utilize a weight vector to quantify preferences for different objectives and incorporate this weight vector as part of the input to the policy network. However, such approaches often struggle with scalability, since for high-dimensional environments the weight space extends exponentially as the number of objectives increases. By comparison, training a restricted set of policies to align with the set of sampled preference vectors can circumvent the scalability issue to some extent. Yet it could be scarcely possible to fulfill the demand of covering the entire Pareto frontier. In addition, although some works focus on boosting sample efficiency (Wiering et al., 2014; Alegre et al., 2023), they may still rely on learning the environment dynamics with extra training time, while inaccurate learned dynamics easily affect multiple RL objectives' performance. Evolutionary approach and user feedback are also integrated to find the Pareto set approximation (Xu et al., 2020; Shao et al., 2024), while additional prediction models are needed to guide the learning process. Although these methods demonstrate promising performance in tasks with simpler dynamics or a limited number of objectives (typically two to five), they often struggle to scale effectively with respect to a greater number of objectives or larger state and action spaces.

In summary, existing works mainly suffer from three aspects: (i) low training efficiency; (ii) hard to cover the complete Pareto front; (iii) inability to maximize utility for any given preference. To address these challenges, we propose a novel constrained optimization formulation for improved computation complexity and stronger coverness of the Pareto front. To be specific, we adapt the constrained policy optimization algorithms (Achiam et al., 2017; Liu et al., 2020), and reformulate the MORL problem with designed constraints on policy performance over multiple objectives. Our approach of training the policy set consists of two stages. In the *Pareto initialization stage*, we train several initial policies in parallel based on fixed preferences until convergence. In the *Pareto extension stage*, we first select diverse policies based on their crowd distance and employ constrained update steps to the initial policies individually. In each optimization step, we optimize a specific objective while constraining the expected returns of other objectives. Through this approach, we can extend the Pareto front in various objective directions. For the third research gap, we introduce *Policy assignment*, which ensures that for any given preference, we assign a policy from the Pareto set that maximizes its utility.

Our algorithm achieves favorable time complexity characteristics and derives a high-quality Pareto front, as indicated by the following observations. Firstly, training several initial policies based on fixed preferences is efficient. In addition, the adoption of constrained update steps allows for rapid adjustment of the initial policy, leading to the derivation of new solutions on the Pareto front. Regarding the second research gap concerning the complete Pareto front, different from the meta-policy approach that relies on a single initial policy (Chen et al., 2019), our method can extend multiple policies selected from initial policies based on their crowd distance to enhance diversity while promoting better performances. Intuitively, a larger crowd distance indicates that the corresponding policy appears on a sparser area on the Pareto front, therefore, extending such a policy is more likely to fill the Pareto front. While sharing similarities with our formulation, as one category of multi-objective optimization (MOO) approach, *epsilon-constraint methods* solves a MOO problem by converting it to several single-objective constrained optimization problems (Laumanns et al., 2006; Van Moffaert & Nowé, 2014). However, the running time of such a method is exponential in problem size, rendering it impractical for MOO problems with numerous objectives. In contrast, crowd-distance-based policy selection eases such burden, exhibiting linear complexity, and can efficiently solve MORL tasks with numerous objectives. Our main contributions are as follows:

- We propose C-MORL, a two-stage policy optimization algorithm that enables rapid and complete discovery of the Pareto front. By taking a novel constrained optimization perspective for MORL, our Pareto front extension method can easily handle complex discrete or continuous MORL tasks with multiple objectives (as demonstrated with up to nine objectives in our experimental evaluations).

- To empirically solve C-MORL without extra computation such as in the epsilon-constraint method, we propose an efficient interior-point-based approach for finding the solution of a relaxed formulation, which can guarantee the derivation of Pareto-optimal policies under specified conditions.

- To validate the efficacy of C-MORL, we employ an array of MORL benchmarks for both continuous and discrete state/action spaces across various domains, such as robotics control and sustainable energy management. C-MORL consistently achieves up to $35\%$ larger hypervolume and $9\%$ higher expected utility in MORL benchmarks than the state-of-the-art baselines, indicating the discovery of broader Pareto front given any preferences.

## 2 RELATED WORK

Prior trials on tackling multi-objective RL fall into training single preference-conditioned policy or multi-policy. Typical single-preference-conditioned policy approaches adopt a policy that takes preferences as part of network inputs and utilizes the weighted-sum scalarization of the value functions (or advantage functions) to optimize the policy (Van Moffaert et al., 2013; Parisi et al., 2016; Yang et al., 2019; Zhang & Golovin, 2020; Basaklar et al., 2022; Lu et al., 2022; Hung et al., 2022; Zhu et al., 2023; Lin et al., 2024). In the evaluation stage, agents can execute corresponding solutions based on users' desired preferences (Yang et al., 2019; Basaklar et al., 2022). With the shared neural networks, gradients from different tasks can interfere negatively, making learning unstable, leading to imbalanced performance across the entire preference space, and sometimes even less data efficient.

Instead of training a single policy agent, multi-policy approaches train a finite set of policies to approximate the Pareto front (Abels et al., 2019; Xu et al., 2020; Alegre et al., 2022). In the evaluation stage, the policy in policy set that maximizes the utility of the user (i.e., the weighted sum of objectives) is chosen (Friedman & Fontaine, 2018). Rather than evenly sampling from preference space, (Xu et al., 2020) proposes an efficient evolutionary learning algorithm to find the Pareto set approximation along with a prediction model for forecasting the expected improvement along vector-valued objectives. Yet the performance highly depends on prediction model's accuracy, and it is challenging to recover the Pareto front due to the long-term local minima issue. This method also suffers from low training efficiency due to the exponential complexity involved in repeatedly calculating the quality of the virtual Pareto front during the training process. (Kyriakis & Deshmukh, 2022) designs a policy gradient solver to search for a direction that is simultaneously an ascent direction for all objectives. (Alegre et al., 2022) firstly trains a set of policies whose successor features form a $\epsilon$-$CCS$ (convex coverage set), then utilizes the generalized policy improvement (GPI) algorithm to derive a solution for a new preference. (Alegre et al., 2023) further introduces a novel Dyna-style MORL method that significantly improves sample efficiency while relying on accurate learning of dynamics. (Röpke et al., 2024) discovers the Pareto front by bounding the search space that could contain value vectors corresponding to Pareto optimal policies using an approximate Pareto oracle, and iteratively removing sections from this space. Our proposed C-MORL distinguishes from their divide-and-conquer scheme by a novel policy selection procedure along with explicitly solving a principled constrained optimization.

Our approach bears connections with (Chen et al., 2019) and (Huang et al., 2022), while distinguishing itself in terms of both the training process and objectives. Compared to the meta-policy approach, adaptation process is not necessary in the evaluation phase of our method. When presented with an unseen preference, the policy in the Pareto front with the highest utility is chosen as the surrogate execution policy. In contrast to (Huang et al., 2022; Kim et al., 2024), we do not aim at solving a constrained RL problem for training the working policy under specific preference. Rather, we propose to leverage the constrained optimization steps to fill the complete Pareto front with enhanced flexibility (Liu et al., 2020; Xu et al., 2021), and design extension and selection algorithms to explicitly promote diverse policies. Constrained optimization techniques are widely adopted to solve multi-objective optimization (MOO) problems. *Epsilon-constraint methods* are a category of MOO techniques that involve pre-defining a virtual grid in the objective space and solving single-objective problems for each grid cell, where the optimum of each problem corresponds to a Pareto-optimal solution (Laumanns et al., 2006). Instead of pre-defining virtual grid, this work proposes crowd distance based policy selection to address the exponential complexity in the epsilon-constraint method.

## 3 PRELIMINARIES

### 3.1 MORL SETUP

In this work, we adopt the general framework of a multi-objective Markov decision process (MOMDP), which is represented by the tuple $< \mathcal{S}, \mathcal{A}, \mathcal{P}, \mathcal{R}_{1:n}, \Omega, f, \gamma >$. Similar to standard MDP, at each timestep $t$, the agent under current state $\mathbf{s}_t \in \mathcal{S}$ takes an action at $\mathbf{a}_t \in \mathcal{A}$, and transits into a new state $\mathbf{s}_{t+1}$ with probability $\mathcal{P}(\mathbf{s}_{t+1}|\mathbf{s}_t, \mathbf{a}_t)$. One notable characteristic of MOMDP is that for $n$ different objectives, the reward is a $n$-dimensional vector $\mathbf{r}_t =$

$[\mathcal{R}_1(\mathbf{s}_t, \mathbf{a}_t), \mathcal{R}_2(\mathbf{s}_t, \mathbf{a}_t), \ldots, \mathcal{R}_n(\mathbf{s}_t, \mathbf{a}_t)] \in \mathbb{R}^n$. For any policy $\pi$, it is associated with a vector of expected return, given as $\boldsymbol{G}^\pi = [G_1^\pi, G_2^\pi, \ldots, G_n^\pi]^\top$, where the expected return of the $i^{th}$ objective is given as $G_i^\pi = \mathbb{E}_{\boldsymbol{a}_{t+1} \sim \pi(\cdot|\boldsymbol{s}_t)}\left[\sum_t \gamma^t \mathcal{R}(\boldsymbol{s}_t, \boldsymbol{a}_t)_i\right]$ for some predefined time horizon. We assume such returns are observable.

The goal of MORL is to find a policy so that each objective's expected return in $\boldsymbol{G}^\pi$ can be optimized. In practice, since training RL typically requires a scalar reward to interact with the training agent, and co-optimizing multiple objectives is hard to achieve an ideal tradeoff, especially in a situation where objectives are competing against each other. To that end, denote $\Omega = \{\boldsymbol{\omega} \in \Omega | \sum_{i=1}^n \omega_i = 1, \omega_i \geq 0\}$ as the preference vector. We use the preference function $f_{\boldsymbol{\omega}}(\mathbf{r})$ to map a reward vector $\mathbf{r}(\mathbf{s}, \mathbf{a})$ to a scalar utility given $\boldsymbol{\omega} : f_{\boldsymbol{\omega}}(\mathbf{r}(\mathbf{s}, \mathbf{a})) = \boldsymbol{\omega}^\top \mathbf{r}(\mathbf{s}, \mathbf{a})$. Our goal is then to find a multi-objective policy $\pi(\mathbf{a}|\mathbf{s}, \boldsymbol{\omega})$ such that the expected scalarized return $\boldsymbol{\omega}^\top \boldsymbol{G}^\pi$ is maximized.

### 3.2 PARETO OPTIMALITY

In MORL, a policy that simultaneously optimizes all objectives does not exist. This holds whenever either of the two objectives is not fully parallel to each other. Thus, a set of non-dominated solutions is desired. We say policy $\pi$ is dominated by policy $\pi'$ when there is no objective under which $\pi'$ is worse than $\pi$, i.e., $G_i^\pi \leq G_i^{\pi'}$ for $\forall i \in [1, 2, \ldots, n]$. A policy $\pi$ is Pareto-optimal if and only if it is not dominated by any other policies. The Pareto set is composed of non-dominated solutions, denoted as $\Pi_P$. The corresponding expected return vector $\boldsymbol{G}^\pi$ of policy $\pi \in \Pi_P$ forms Pareto front $P$. In this paper, an element in solution set $\mathcal{X}_P$ refers to a policy along with its corresponding expected return vector $(\pi, \boldsymbol{G}^\pi)$. However, in many complex real-world control problems, obtaining the optimal Pareto set is challenging, and it is becoming even more difficult considering the sequential decision-making nature of RL problems. Thus the goal of MORL is to obtain a Pareto set $P$ to approximate the optimal Pareto front.

With a finite Pareto set, we can define the Set Max Policy (SMP) (Zahavy et al., 2021) of a given preference vector:

**Definition 3.1.** (Set Max Policy). Denote $\Pi_P$ to be a Pareto set. Then, Set Max Policy (SMP) is the best policy in the set $\Pi_P$ for a given preference vector $\boldsymbol{\omega}$: $\pi_{\boldsymbol{\omega}}^{SMP} = \max_{\pi \in \Pi_P} f_{\boldsymbol{\omega}}(\boldsymbol{G}^\pi)$.

The notion of SMP enables the direct assignment of a surrogate execution policy given an unseen preference in the evaluation phase. Fig. 1 illustrates the determination of SMP.

To evaluate a Pareto set $\Pi_P$, there are three evaluation metrics introduced to compare the quality of the Pareto front and utility achieved by the underlying algorithm: (i) hypervolume, (ii) expected utility, and (iii) sparsity (Hayes et al., 2022). The details of these evaluation metrics are provided in Appendix E.3.

**Definition 3.2.** (Crowd Distance). Let $P$ be a Pareto front approximation in an $n$-dimensional objective space. Also denote $\tilde{G}_i$ as the ascending sorted list with a length of $|P|$ for the $i^{th}$ objective values in $P$. Given the $j^{th}$ solution, and suppose the sort sequence of $P(j)$ in $\tilde{G}_i$ is $k$, then the Crowd Distance of solution $P(j)$ is $D(P(j)) = \sum_{i=1}^n \frac{\tilde{G}_i(k+1) - \tilde{G}_i(k-1)}{\tilde{G}_{i,max} - \tilde{G}_{i,min}}$.

The crowd distance is a measure of how close an individual is to its neighbors (Deb et al., 2002).

## 4 OPTIMIZING CONSTRAINED MORL

In this section, we start from converting the MORL problem as a constrained MDP (CMDP) while guaranteeing the local optimality under such formulation. Next, we present the conditions under which a feasible solution to the CMDP problem qualifies as a Pareto-optimal solution. Then, to effectively solve the CMDP problem, we prove that under primal-dual formulation, despite its non-convexity, the CMDP problem has zero duality gap, i.e., it can be solved exactly in the dual domain, where it becomes convex (Paternain et al., 2019).

Constrained RL problems are typically formulated as constrained MDPs (CMDP) (Altman, 2021). A

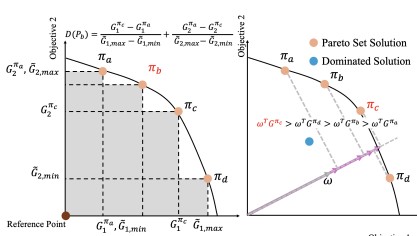

Figure 1: Visualization of metrics. (a) Hypervolume, reference point, and example of crowd distance calculation. As an example, the crowd distance of $\pi_b$ is calculated based on the expected return of its neighbors $\pi_a$ and $\pi_c$, as well as the extreme solutions on the two objectives. (b) Given a preference vector, the corresponding expected return is calculated by selecting the set max policy from Pareto front solutions.

CMDP is defined by the tuple $< \mathcal{S}, \mathcal{A}, \mathcal{P}, \mathcal{R}_{1:n}, \mathbf{d}_{1:n}, \gamma >$. The reward function of CMDP includes the reward function $\mathcal{R}_l(\mathbf{s}_t, \mathbf{a}_t)$ of the $l^{th}$ objective that is being optimized, and the constraint-specific reward functions $\{\mathcal{R}_i(\mathbf{s}_t, \mathbf{a}_t)\}_{(1:n)\setminus l}$ of other objectives. $\mathbf{d} := \{d_i\}_{(1:n)\setminus l}$ represents the corresponding thresholds of constraint-specific reward functions.

The optimal solution to a CMDP is a policy that maximizes expected return of the objective being optimized, while ensuring the expected returns of other constraints satisfy their baseline thresholds. We adapt to the following constrained RL formulation to ensure when we optimize $\pi$ for $G_l^\pi$, the returns of other objectives are not seriously hampered:

$$\max_\pi G_l^\pi \quad \text{s.t. } G_i^\pi \geq d_i \quad i = 1, \ldots, n, \ i \neq l. \tag{1}$$

**Assumption 4.1.** The value $G_i^\pi(s), i = 1, ..., n, \ i \neq l$ is bounded for all policies $\pi \in \Pi_P$.

**Assumption 4.2.** Every local minima of $G_l^\pi$ is a feasible solution.
Assumption 4.2 justifies the use of gradient-based algorithm for solving CMDP can converge with general constraints. As long as the threshold $\mathbf{d}$ are chosen beforehand such that for initializations, there are $G_i^\pi \geq d_i, i \neq l$, then starting points for problem Eq. 1 is always feasible.

Note that implementing the Pareto extension aims to expand the Pareto front in various objective directions, starting from the initialization points of problem Eq. 1. In this regard, the solution derived from Eq. 1 should contribute to the Pareto front, meaning it must be a Pareto-optimal solution. Therefore, it is crucial to set proper constraint values $d_i, i = 1, ..., n, \ i \neq l$. Accordingly, we present the following proposition that formalizes the criteria for specifying appropriate constraint values:

**Proposition 4.3.** *Let $\tilde{G}_i$ denote the ascending sorted list for the $i^{th}$ objective values in P, and suppose the sorted sequence of the initial point $P_r$ in $\tilde{P}_i$ is k. If $d_i \geq \tilde{G}_i(k-1)$ for all $i = 1, \ldots, n, \ i \neq l$, then the optimal solution of problem Eq. 1 is a Pareto-optimal solution.*
See Proof in Appendix B. Proposition 4.3 provides the condition of which the feasible solution of problem Eq. 1 is a Pareto optimal solution. Fig. 2 visualizes the constraint value setting criteria. In the next section, we will further propose a more practical method of specifying constraint values $d_i$ for all $i = 1, \ldots, n, \ i \neq l$.

Having established the criteria for selecting appropriate constraints, we now turn our attention to the challenges associated with solving the constrained formulation Eq. 1, which is untractable due to the nonconvex and multiple-step formulation. To address such issue, constrained CMDP formulation of MORL problem Eq. 1 can be solved by using the Lagrangian relaxation technique once converted to an unconstrained problem. Denote $\boldsymbol{G}_{1:n\setminus l}^\pi = [G_1^\pi, ..., G_n^\pi]^\top, i \neq l$. And define the Lagrangian $L := G_l^\pi - \boldsymbol{\lambda}^\top (\mathbf{d} - \boldsymbol{G}_{1:n\setminus l}^\pi)$, where $\lambda_i \geq 0, i = 1, ..., n, \ i \neq l$ is the Lagrange multiplier. The resulting equivalent problem is the dual problem

$$D^* \triangleq \min_{\boldsymbol{\lambda}} \max_\pi \ L(\boldsymbol{\lambda}, \pi). \tag{2}$$

Denote the projection operator as $\Gamma_{\boldsymbol{\lambda}}, \Gamma_\pi$, which projects $\boldsymbol{\lambda}$ and $\pi$ to compact and convex sets respectively. It can be shown that at optimization iteration step $r$, for our CMDP formulation of the underlying MORL problem, iteratively working on $\boldsymbol{\lambda}_r, \pi_r$ with step size $\eta_1, \eta_2$ is guaranteed to reach the local minima of the unconstrained problem Eq. 2 (Borkar, 2009; Tessler et al., 2018):

**Proposition 4.4.** *Under mild conditions and by implementing the update rules as followed for $\boldsymbol{\lambda}$ and $\pi$: $\lambda_{r+1,i} = \Gamma_\lambda[\lambda_{r,i} - \eta_1(r)\nabla_\lambda L(\lambda_{r,i}, \pi_r)]; \quad \pi_{r+1} = \Gamma_\pi[\pi_r - \eta_2(r)\nabla_\pi L(\boldsymbol{\lambda}_r, \pi_r)];$ such iterates $(\boldsymbol{\lambda}_n, \pi_n)$ converge to a fixed point of MORL policy almost surely.*
See proof in Appendix C. Indeed, (Paternain et al., 2019) shows that there is zero duality gap between the original CMDP formulation and its dual formulation. Motivated by such property, if we denote the optimal solution of the original MORL problem Eq. 1 as $P^*$, we can alternatively work on finding the solution of the dual formulation Eq. 2 to derive the optimal MORL policy.

Further, to justify the use of interior point method adopted by our C-MORL-IPO described in Section 4, the following Theorem helps connect the solution procedure via the log barrier method with the optimal solution of the original C-MORL problem:

**Theorem 4.5.** *Suppose that $\mathcal{R}_i$ is bounded for all $i = 1, \ldots, n$ and that Slater's condition holds for problem Eq. 1. Then, strong duality holds for 1, i.e., $P^* = D^*$. Define the logarithmic barrier function for each constraint $\phi(G_i^\pi) = \frac{\log(G_i^\pi - \beta G_i^{\pi_r})}{t}$ with $t$ as a hyperparameter. If the optimal policy for C-MORL is strictly feasible, the maximum gap between Eq. 1 and solving it via the interior point method is bounded by $\frac{n-1}{t}$.*

The above Theorem indicates that we can safely solve C-MORL problem with tunable parameter $t$ on the log barrier. With such properties, we are ready to show our design strategies to more efficiently discover the Pareto front via solving C-MORL involving multiple constraints on the return of heterogeneous preferences.

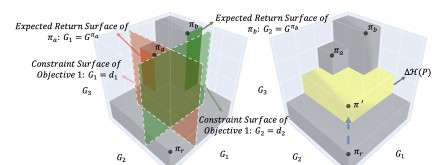

# 5 CONSTRAINED MORL-BASED PARETO FRONT EXTENSION

In this section, we introduce our two stage-design for the construction of the Pareto set via the proposed C-MORL along with the selection scheme for the second extension stage.

Figure 2: Visualization of criteria for specifying constraint values. $\pi_r$ denotes initial point. The expected return $G^{\pi_a}(G^{\pi_b})$ of solution $P_a(P_b)$ in objective 1(2) is the $(k-1)^{th}$ value in list $\tilde{G}_1(\tilde{G}_2)$, respectively. Therefore, specifying constraints values $d_1 \geq G^{\pi_a}$ and $d_2 \geq G^{\pi_b}$ is sufficient for the feasible solution of corresponding Eq. 1 to be Pareto-optimal solution.

## 5.1 PARETO INITIALIZATION

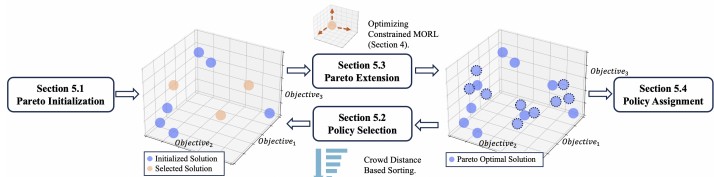

Figure 3: Procedure of two-stage C-MORL. *Pareto initialization*: training several initial policies to derive the initial solution set $\mathcal{X}_{init}$. *Pareto extension*: iteratively implementing policy selection and Pareto extension with constrained policy optimization toward desired Pareto extension directions in the objective space. *Policy assignment*: given preference $\boldsymbol{\omega}$, the surrogate execution policy selected from the Pareto set based on Eq. 3.1.

As shown in Fig. 3, during the Pareto initialization stage, we first train and collect a set of initial policies. Each policy $\pi$ corresponds to a pre-known preference vector $\boldsymbol{\omega}$, as we train policy $\pi$ using the multi-objective policy gradient algorithm (Xu et al., 2020) to maximize the return under this particular $\boldsymbol{\omega}$. Specifically, we also extend the value function to a vectorized version with vectorized target value $\hat{\mathbf{V}}(\mathbf{s})$. Then in the policy update iterations, we utilize the vectorized advantage function $\mathbf{A}^\pi(\mathbf{s}_t, \mathbf{a}_t)$ to implement policy gradient updates: $\nabla_\theta \mathcal{J}(\theta, \boldsymbol{\omega}) = \sum_{i=1}^n \omega_i \nabla_\theta J_i(\theta) = \mathbb{E}\left[\sum_{t=0}^T \boldsymbol{\omega}^\top \mathbf{A}^\pi(\mathbf{s}_t, \mathbf{a}_t) \nabla_\theta \log \pi_\theta(\mathbf{a}_t|\mathbf{s}_t)\right].$

By sampling diverse preference vectors to guide the training of initial policies, we obtain an initial solution set $\mathcal{X}_{init}$. It is important to note that the preference vector is solely used for guiding the training of these initial policies, and the resulting solutions are preference-irrelevant, meaning the initialized policies can be further trained without specifying preference. Any policy in the set $\mathcal{X}_{init}$ can also be assigned to and evaluated by any new preference in the policy selection stage. Note that in the Pareto initialization stage, we also maintain a solution buffer to enhance policy diversity.

## 5.2 POLICY SELECTION

Direct implementation of the Pareto extension based on policies from $\mathcal{X}_{init}$ could encounter several limitations. First, some policies may not lie on the Pareto front, making their extension less effective for discovering new Pareto-optimal solutions. Furthermore, the distribution of Pareto-optimal policies along the Pareto front may be uneven. Random selection for extension could result in trajectory overlap and subsequent inefficiencies, or leaving regions of the Pareto front unexplored. Inspired by multi-objective optimization algorithms (Deb et al., 2002), we design policy selection schemes before and during the Pareto extension stage respectively.

The Pareto-optimal policies are selected based on their crowd distance. A larger crowd distance indicates that the corresponding region on the Pareto front is relatively unexplored, and therefore, it may hold a greater potential for augmenting the Pareto front during the next iteration of the Pareto extension. We sort all Pareto-optimal policies according to their crowd distance value and select the top $N$ policies with the greatest crowding distance to construct $\mathcal{X}_{extension}$ for the subsequent Pareto extension iteration. C-MORL selects policies after the Pareto initialization stage and every $\frac{K}{K'}$ steps during the Pareto extension stage, where $K$ is the total number of Pareto extension step, $K'$ is the number of constrained optimization steps between two iterations of policy selection. The detailed policy selection procedure is summarized as Algorithm 1 in Appendix A.

## 5.3 PARETO EXTENSION

In this Pareto extension stage, we achieve the goal of filling the Pareto front from selected solutions $\mathcal{X}_{extension}$ toward different directions by solving constrained optimization on the selected policies, as is shown in the Pareto Extension stage in Fig. 3. In each constrained optimization step, we optimize one of the objectives, denoted as $l$. By performing such optimization on the initial policy for all listed objectives, we are able to obtain a set of policies that approximate the Pareto front towards various directions while fully utilizing currently collected initial policies.

In Section 4, we present Proposition 4.3 as a sufficient condition for specifying constraint values that ensures the feasible solution of Eq. 1 is a Pareto-optimal solution. However, this Proposition is impractical for several reasons. First, the condition is too strict and may lead to the exclusion of discovering some potential Pareto-optimal solutions. Second, it necessitates evaluating the expected return of all policies for non-dominated sorting at each step of the constrained optimization, which is inefficient. Therefore, we propose an alternative constraint specification method that utilizes only the expected return of the policy from the most recent step. Specifically, we consider solving the following problem in which return constraints are controlled by a hyperparameter $\beta \in (0,1)$:

$$\pi_{r+1,i} = \arg \max_{\pi \in \Pi_\theta} \{G_l^\pi : G_i^\pi \geq \beta G_i^{\pi_r}, i = 1, \ldots, n, i \neq l\}, \tag{3}$$

where $G_i^{\pi_r}$ is the expected return of the $i^{th}$ objective in the last constrained optimization step, which is indexed by $r$ for the current iteration. Next, we introduce practical methods to solve Eq. 3.

**C-MORL-IPO** Constrained Policy Optimization (CPO) is a widely used general-purpose policy search algorithm for constrained RL (Achiam et al., 2017). While the CPO algorithm ensures monotonic policy improvement and guarantees constraint satisfaction throughout the training process, inner-loop optimization is required when there are multiple constraints. Therefore, CPO is hard to handle more than two MORL objectives. To overcome this issue, inspired by (Xu et al., 2021; Liu et al., 2020), we also propose to find a satisfiable solution efficiently for Eq. 3 via the interior point method (IPO), which is a primal type approach and holds the promise of finding a solution closer to the original C-MORL as shown in Theorem 4.5. Given $G_i^{\pi_r}$ and the the defined log barrier in 4.5, we then convert problem Eq. 1 into an unconstrained optimization problem:

$$\max_\pi \ G_l^\pi + \sum_{i=1}^n \phi(G_i^\pi) \tag{4}$$

The detailed procedures of solving Eq.4 are provided in Appendix F. In practice, IPO is more robust in stochastic environments, and larger $t$ would guide to a solution with higher rewards yet with more computation costs. Without the requirement of calculating the second-order derivative, C-MORL-IPO is more computationally efficient than trust-region-based method for CPO updates (Achiam et al., 2017), which we term as C-MORL-CPO and compare it in Appendix F.

We note that C-MORL can achieve a better approximation of the Pareto front by design, which is different from previous CMDP approaches (Abdolmaleki et al., 2020; Huang et al., 2022). Such type of methods typically have an explicit learning- or optimization-based policy improvement stage to solve for particular preferences. Implementing C-MORL helps directly move away from dominated solutions in the Pareto front, so that the hypervolume can be directly optimized, while resulting policies are also more generalizable. The following proposition analyzes the time complexity of C-MORL, demonstrating its linear time complexity with respect to the number of objectives, making it suitable for MORL tasks with high-dimensional objective spaces:

**Proposition 5.1.** *(Time complexity.) Given that the number of objectives is $n$, the number of extension policies is $N$, assume the running time of each optimization problem is upper bounded by a constant, and the number of Pareto extension steps is $K$, the expected running time of Algorithm 2 is $O(nKN)$.*

Compared to PG-MORL (Xu et al., 2020) which needs to solve a knapsack problem with $K \times N$ candidate points to evaluate and need $O(KN^{n-2})$ steps to solve it exactly, the proposed C-MORL excels when there are growing number of objectives or steps.

## 5.4 PARETO REPRESENTATION VIA POLICY ASSIGNMENT

**Policy Assignment** During the Pareto extension stage, we store the policies with their corresponding expected return on the Pareto front. Therefore, after the Pareto extension stage, we derive an approximated Pareto front with Pareto set policies, as shown in the Policy assignment process in Fig. 3. With Pareto set policies, given an unseen preference, we can select its SMP $\pi_{\boldsymbol{\omega}}^{SMP}$ by solving

Eq. 3.1. Such a policy assignment process is efficient and does not require any retraining of the new policy. Algorithm 2 in Appendix A presents the complete workflow of C-MORL.

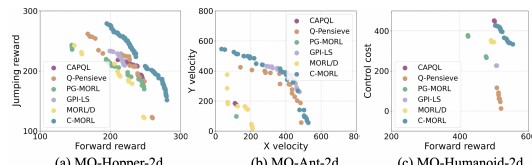

Figure 4: Pareto front comparison on two-objective MO-MuJoCo benchmarks.

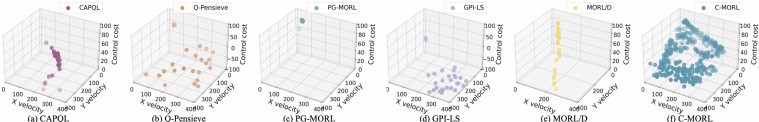

Figure 5: Pareto front comparison on MO-Ant-3d benchmark.

## 6 EXPERIMENTS

In this Section, we validate the design of our proposed algorithm using both popular discrete and continuous MORL benchmarks from MO-Gymnasium (Felten et al., 2023a) and SustainGym (Yeh et al., 2024). These benchmarks include five comprehensive domains: (i) **Grid World** includes Fruit-Tree, a discrete benchmark with six objectives. (ii) **Classic Control** includes MO-Lunar-Lander, a discrete benchmark with four objectives. (iii) **Miscellaneous** includes Minecart, a discrete benchmark with four objectives. (iv) **Robotics Control** includes five MuJoCo tasks with continuous action space based on MuJoCo simulator (Todorov et al., 2012; Xu et al., 2020). (v) **Sustainable Energy Systems** includes two building heating supply tasks. These benchmarks pose significant challenges for MORL due to complex state (up to $348$)/action (up to $23$) spaces and large objectives spaces (up to 9 objectives). More details of experiment settings are listed in Appendix E.

We benchmark our algorithm against five competitive baselines under the metrics of hypervolume (HV), expected utility (EU), and sparsity metrics (SP). Higher HV and EU, lower SP indicate better performance. Each of the baselines are trained for $5 \times 10^5$ time steps for discrete benchmarks. Continuous benchmarks with two, three, and nine objectives are trained for $1.5 \times 10^6$, $2 \times 10^6$, and $2.5 \times 10^6$ steps, respectively. The baselines involve: (i)**Envelope** (Yang et al., 2019).(ii) **CAPQL** (Lu et al., 2022). (iii)**Q-Pensieve** (Hung et al., 2022). (iv) **PG-MORL** (Xu et al., 2020). (v) **GPI-LS** (Alegre et al., 2023). (vi) **MORL/D** (Felten et al., 2024). Among these baselines, **Envelope** is developed for discrete control tasks, **GPI-LS** is suitable for both discrete and continuous control. The other baselines are specifically tailored for continuous control.

**Overall Results.** To assess the performance of MORL, we begin by comparing the quality of the Pareto front. It can be observed from Table 1 and Table 2 that the proposed C-MORL achieves the highest hypervolume in all benchmarks. This indicates that C-MORL successfully discovers a high-quality Pareto front across various domains, particularly in benchmarks with large state and action spaces (MO-Humanoid-2d) and a substantial number of objectives (Building-9d). Figures 4 and 5 illustrate the Pareto front for MO-MuJoCo benchmarks with two and three objectives, respectively. C-MORL exhibits more comprehensive coverage of the Pareto front across all benchmarks, whereas other baselines fail to encompass the entire front. For instance, in the MO-Ant-2d benchmark, Q-Pensieve does not cover the upper-left portion of the Pareto front, indicating insufficient exploration of the Y velocity objective, which results in lower utility for preference pairs prioritizing this objective.

Table 1: Evaluation of HV, EU, and SP for discrete MORL tasks.

| Environments | Metrics | Envelope | GPI-LS | C-MORL |
|---|---|---|---|---|
| Minecart | HV($10^2$) | 1.99±0.00 | **6.05±0.37** | **6.77±0.88** |
| | EU($10^{-1}$) | -2.72±1.01 | **2.29±0.32** | **2.12±0.66** |
| | SP($10^{-1}$) | 5.11±2.11 | 0.10±0.00 | **0.05±0.02** |
| MO-Lunar-Lander | HV($10^9$) | 0.43±0.18 | 1.06±0.16 | **1.12±0.03** |
| | EU($10^1$) | -2.84±4.06 | 1.81±0.34 | **2.35±0.18** |
| | SP($10^3$) | 0.19±0.16 | **0.13±0.01** | 1.04±1.24 |
| Fruit-Tree | HV($10^4$) | **3.66±0.23** | 3.57±0.05 | **3.67±0.14** |
| | EU | 6.15±0.00 | 6.15±0.00 | **6.53±0.03** |
| | SP($10^{-1}$) | 5.46±0.15 | 5.29±0.21 | **0.42±0.03** |

While C-MORL leads the highest hypervolume by a large margin in almost all cases, it does not always attain the best sparsity in some benchmarks. This occurs because, in certain cases, an

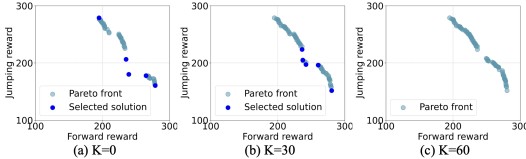

Figure 6: Pareto extension stage of MO-Ant-2d benchmark. The blue points are Pareto-optimal solutions, and the deep-blue points are selected solutions.

algorithm may identify only a few similar Pareto-optimal solutions, leading to low sparsity. As illustrated in Table 2, along with Figures 4 (c) and 5, both CAPQL and GPI-LS cover only a specific portion of the Pareto front, resulting in a reduced sparsity. In contrast, C-MORL generates a dense and comprehensive Pareto front in these benchmarks. C-MORL also demonstrates the highest expected utility in nine of the ten benchmarks, which indicates that C-MORL can better maximize user utility given various user preferences. Appendix G.3 further illustrates that as a multi-policy approach, C-MORL can align corresponding Pareto-optimal solution given any preference. This is in contrast to single preference-conditioned policies, which may yield dominated solutions for unseen preferences.

Regarding time complexity, as stated in Section 4, C-MORL is linear time complexity with respect to the number of objectives, and is therefore generalizable to benchmarks with more than three objectives. We also note from Table 2 that for the Building-9d benchmark with nine objectives, the training times for PG-MORL and GPI-LS are excessively long, exceeding the time limitation we set. This finding aligns with the time complexity analysis of PG-MORL discussed in Section 4. Additionally, in both the Pareto initialization and extension stages, the policies can be trained in parallel, which further reduces the total running time. Given that the expected running time is $O(nKN)$, the Pareto extension steps $K$ and the number of selected policies $N$ can be adjusted to balance performance with running time.

**Pareto extension analysis.** In Fig. 6, we showcase how the Pareto front is filled using a set of selected policies with our Pareto extension approach. During the initialization stage of the Pareto set, we uniformly sample preference pairs from the preference space. Training a single policy with a fixed preference vector facilitates convergence to a Pareto-optimal solution. Additionally, we present an ablation study on the number of selected extension policies in Appendix G.1.

Building on this foundation, the subsequent Pareto set extension stage further populates the Pareto front. As shown in Fig. 6, gaps remain on the Pareto front following the initialization stage. Our effective policy selection algorithm targets Pareto-optimal solutions in these gap regions, which exhibit higher crowding distances, thereby selecting them for the ensuing extension stage. During the Pareto extension phase, as the number of extension steps $K$ increases, C-MORL employs constrained policy optimization to adjust the selected policies toward various objective directions, effectively filling the gaps in the Pareto front.

Next, to better understand the influence of hyperparameters and key components of C-MORL, we perform an in-depth analysis and conduct ablation studies focusing on these aspects.

**Parameter study of C-MORL for Pareto Initialization.** As mentioned in Section 5.1, during Pareto initialization stage, C-MORL aims to derive a few Pareto optimal solutions by training a set of initial policies. To study the impact of the number of initial policies $M$, we conduct experiments while keeping the total number of training steps fixed at $1.5 \times$

Table 3: Parameter study of C-MORL for Policy Initialization on MO-Hopper-2d benchmark.

| | HV($10^5$) | EU($10^2$) | SP($10^2$) |
|---|---|---|---|
| M=3 (1.5M steps) | 1.38±0.12 | 2.53±0.13 | 0.58± 0.43 |
| M=6 (1.5M steps) | 1.39±0.04 | 2.55±0.04 | **0.16±0.11** |
| M=11 (1.5M steps) | 1.32±0.03 | 2.47±0.03 | 0.44±0.33 |
| M=6 (3M steps) | **1.45±0.05** | **2.63±0.05** | 0.22±0.10 |

$10^6$ steps (including $1 \times 10^6$ steps for initialization stage) to ensure a fair comparison. Specifically, we uniformly sample $M = 3, 6, 11$ preference vectors, as shown in Table 3. For example, with a sampling interval of 0.2, the preference vectors are $\boldsymbol{\omega} = [0, 1], [0.2, 0.8], [0.4, 0.6], [0.6, 0.4], [0.8, 0.2], [1, 0]$. Intuitively, increasing $M$ can enhance diversity among Pareto-optimal solutions, which benefits the subsequent extension phase. However, since the number of total time steps is fixed, increasing $M$ reduces the training steps allocated to each initial policy. The trade-off is evident as the performance for $M = 11$ is worse than for $M = 6$, indicating that the increasing of $M$ does not always guarantee

Table 2: Evaluation of HV, EU, and SP for continuous MORL tasks. *T/O* indicates that the training time exceeded the maximum limit of 100 hours.

| Environments | Metrics | CAPQL | Q-Pensieve | PG-MORL | GPI-LS | MORL/D | C-MORL |
|---|---|---|---|---|---|---|---|
| MO-Hopper-2d | HV($10^5$) | 1.15±0.08 | 1.26±0.01 | 1.20±0.09 | 1.19±0.10 | 1.11±0.03 | **1.39±0.01** |
| | EU($10^2$) | 2.28±0.07 | 2.28±0.01 | 2.34±0.10 | 2.33±0.10 | 2.19±0.04 | **2.56±0.02** |
| | SP($10^2$) | **0.46±0.10** | 1.61±1.31 | 5.13±5.81 | 0.49±0.37 | 2.72±2.05 | 0.33±0.28 |
| MO-Hopper-3d | HV($10^7$) | 1.65±0.45 | 1.66±1.20 | 1.59±0.45 | 1.70±0.29 | 1.94±0.05 | **2.29±0.01** |
| | EU($10^2$) | 1.53±0.28 | 1.26±0.79 | 1.47±0.25 | 1.62±0.10 | 1.68±0.02 | **1.80±0.01** |
| | SP($10^2$) | 2.31±3.16 | 1.77±0.88 | 0.76±0.91 | 0.74±1.22 | 0.84±0.17 | **0.28±0.09** |
| MO-Ant-2d | HV($10^5$) | 1.11±0.69 | 2.55±0.54 | 0.35±0.08 | 3.10±0.25 | 1.03±0.26 | **3.13±0.20** |
| | EU($10^2$) | 2.16±0.94 | 3.14±0.49 | 0.81±0.23 | 4.28±0.19 | 2.22±0.49 | **4.29±0.19** |
| | SP($10^3$) | **0.18±0.07** | 3.63±2.71 | 2.20±3.48 | 3.61±2.13 | 2.90±2.61 | 1.67±0.85 |
| MO-Ant-3d | HV($10^7$) | 1.22±0.33 | 3.82±0.43 | 0.94±0.12 | 0.55±0.81 | 1.00±0.17 | **4.09±0.13** |
| | EU($10^2$) | 1.30±0.29 | 2.18±0.41 | 1.07±0.07 | 2.41±0.20 | 1.51±0.12 | **2.57±0.06** |
| | SP($10^3$) | 0.17±0.09 | 0.83±0.07 | **0.02±0.01** | 1.96±0.79 | 0.85±0.57 | 0.03±0.01 |
| MO-Humanoid-2d | HV($10^5$) | 3.30±0.05 | 0.90±0.62 | 2.62±0.32 | 1.98±0.02 | 2.81±0.07 | **3.43±0.06** |
| | EU($10^2$) | **4.75±0.04** | 1.89±0.45 | 4.06±0.32 | 3.67±0.02 | 4.32±0.06 | **4.78±0.05** |
| | SP($10^4$) | **0.00±0.00** | 1.08±1.39 | 0.13±0.17 | 0.00±0.00 | 0.00±0.00 | 0.18±0.27 |
| Building-3d | HV($10^{12}$) | 0.33±0.18 | 1.00±0.02 | 0.83±0.02 | 0.26±0.04 | 0.87±0.38 | **1.15±0.00** |
| | EU($10^4$) | 0.75±0.09 | 0.96±0.00 | 0.93±0.01 | 0.74±0.01 | 0.95±0.00 | **1.02±0.00** |
| | SP($10^5$) | 0.18±0.08 | 0.92±0.78 | **0.04±0.02** | 0.07±0.09 | 7.31±2.20 | 0.07±0.06 |
| Building-9d | HV($10^{31}$) | 4.29±0.73 | 7.28±0.57 | T/O | T/O | T/O | **7.93±0.07** |
| | EU($10^3$) | 3.31±0.06 | 3.46±0.03 | T/O | T/O | T/O | **3.50±0.00** |
| | SP($10^3$) | 4.34±3.72 | **1.04±0.38** | T/O | T/O | T/O | 2.79±0.40 |

better performance. To further investigate, we increase the number of training steps to $3 \times 10^6$ (with $2 \times 10^6$ steps allocated to initialization stage) while keeping $M = 6$. The results demonstrate that proportionally increasing the total training steps and the number of initial policies leads to improved performance, highlighting the importance of balancing training resources with policy diversity.

**Parameter Study for Return Constraint Hyperparameter $\beta$.** In Section 5.3 and Section 4, we develop criteria and practical methods for specifying constraint values for problem Eq. 1. We conduct parameter study for return constraint hyperparameter $\beta$ in Eq. 3 to examine how constraint values influence the generation of Pareto-optimal solutions. Table 4 presents the results of C-MORL with various $\beta$ on the MO-Ant-2d benchmark. When $\beta = 0.9$, C-MORL presents the best performance on all metrics. Intuitively, a higher $\beta$ ensures that when optimizing a specific objective, the expected returns on other objectives do not decrease significantly, thereby facilitating adaptation to new Pareto-optimal solutions.

**Ablation Study on Policy Selection.** To evaluate the effectiveness of the crowding distance-based policy selection method, we compare the Pareto extension using this approach with random selection. Table 5 presents the comparison results, where *Random* and *Crowd* refer to random policy selection and crowd distance based

Table 4: Evaluation of HV, EU, and SP for MO-Ant-2d benchmark with different $\beta$.

| $\beta$ | HV($10^5$) | EU($10^2$) | SP($10^3$) |
|---|---|---|---|
| 0.5 | 3.05±0.26 | 4.24±0.25 | 2.54± 2.41 |
| 0.7 | 3.07±0.26 | 4.23±0.24 | 2.06±1.51 |
| 0.9 | **3.08±0.25** | **4.27±0.22** | **1.71±0.21** |

policy selection. Each method is tested across three runs with the same seeds to ensure consistent Pareto initialization and uses the same hyperparameters for the Pareto extension. The experimental results show that C-MORL with crowd distance based policy selection outperforms the random selection across all metrics, highlighting the effectiveness of the proposed policy selection method. Intuitively, adjusting the policies in sparse areas facilitates better filling of the gaps in the Pareto front.

## 7 CONCLUSIONS AND DISCUSSIONS

This paper introduces a novel formulation and efficient solution procedure for multi-objective reinforcement learning problems. It leverages a two-stage approach to construct the Pareto front efficiently. By em-

Table 5: Evaluation of HV, EU, and SP for MO-Ant-3d benchmark with different policy selection methods.

| Selection method | HV($10^7$) | EU($10^2$) | SP |
|---|---|---|---|
| Random | 3.91±0.31 | 2.51±0.13 | 8.69±5.02 |
| Crowd | **4.10±0.14** | **2.57±0.05** | **8.34±2.84** |

ploying a precise selection of sparse regions on the Pareto front that require further exploration and the utilization of a customized extension method, C-MORL not only achieves superior performance in terms of training efficiency but also excels in terms of both Pareto front quality and user utility compared to state-of-the-art baselines on both discrete and continuous benchmarks. In future work, we plan to develop a more effective extension method so as to more efficiently discover the unexplored Pareto front, especially for continuous environments. We are also interested in designing transfer schemes of learned MORL policies, as well as coordination of agents for mastering real-world MORL tasks with agent-specific reward signals.

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

# A  POLICY SELECTION AND C-MORL ALGORITHM

This section provides details of policy selection and C-MORL algorithms.

In our C-MORL implementation, during the policy initialization stage, we employ parallel training of $M$ policies for pre-known preference vector $\omega$. We maintain a policy buffer, meaning that in addition to the final policy obtained in the Pareto initialization stage, intermediate policies are also saved in the buffer. As a result, the number of candidate policies in the policy selection stage is usually larger than $M$, incorporating the policies stored in the solution buffer.

We conduct policy selection in both after the Pareto initialization stage and during the Pareto extension stage. Specifically, the number of extension policies $N$, the number of Pareto extension steps $K$ and the number of constrained optimization steps $K^{'}$ are predefined. During the Pareto extension stage, C-MORL selects Pareto optimal solutions based on their crowd distance every $\frac{K}{K'}$ steps to make sure that the sparse areas on Pareto-front have a higher potential to be explored.

Furthermore, in the policy selection process, the extreme policies on the Pareto front (for each objective, the extreme solution on the Pareto front refers to the solution that achieves the maximum value for that particular objective) are selected by default. The other Pareto-optimal solutions are selected based on their crowd distance. The policy selection process continues until a predetermined number of $N$ policies are selected or until all Pareto-optimal policies have been selected. This procedure is detailed in the following Algorithm 1.

---

**Algorithm 1** Policy Selection

---

**Require:** Number of extension polices $N$, solution set $\mathcal{X}$.
 1: Initialize extension solution set $\mathcal{X}_{extension} = \{\}$, number of selected policies $N_{selected} = 0$.
 2: Solution set $\mathcal{X} \leftarrow$ filter Pareto-optimal solutions in solution set $\mathcal{X}$.
 3: Sort solutions in solution set $\mathcal{X}$ by crowd distance in descending order.
 4: **while** $N_{selected} < N$ **and** $N_{selected} \neq |\mathcal{X}|$ **do**
 5:     Add the $N_{selected}^{th}$ solution in solution set $\mathcal{X}$ into extension solution set $\mathcal{X}_{extension}$.
 6:     $N_{selected} = N_{selected} + 1$
 7: **end while**
**Ensure:** extension solution set $\mathcal{X}_{extension}$.

---

In Algorithm 2, we describe the whole procedure of our proposed C-MORL algorithm.

---

**Algorithm 2** C-MORL

---

**Require:** Number of initial polices $M$, Number of extension polices $N$,
    initial policy task set $\mathcal{I} = \{(\boldsymbol{\omega}_{init}, \pi_{init})\}_{init=1}^{M}$, solution set $\mathcal{X} = \{\}$,
    initial solution set $\mathcal{X}_{init} = \{\}$, number of objectives $n$, number of Pareto extension step $K$,
    number of constrained update steps $K^{'}$, preference set $\Omega$, number of evaluation preferences $E$.
 1: \\Pareto Initialization.
 2: **for** $(\boldsymbol{\omega}_{init}, \pi_{init}) \in \mathcal{I}$ **do**
 3:     Solution $(\pi_{init}, \boldsymbol{G}^{\pi_{init}}) \leftarrow$ solve task $(\boldsymbol{\omega}_{init}, \pi_{init})$.
 4:     Store solution $(\pi_{init}, \boldsymbol{G}^{\pi_{init}})$ in solution set $\mathcal{X}_{init}$
 5: **end for**
 6: \\Policy Selection. (Algorithm 1)
 7: $\mathcal{X}_{extension} \leftarrow PolicySelection(N, \mathcal{X}_{init})$
 8: \\Pareto Extension.
 9: **for** $iter = 1, \ldots, K/K^{'}$ **do**
10:     \\Constrained Policy Optimization.
11:     **for** $(\pi, \boldsymbol{G}^{\pi}) \in \mathcal{X}_{extension}$ **do**
12:         **for** $i = 1, \ldots, n$ **do**
13:             Initialize $\pi_{i,0} = \pi$.
14:             **for** $r = 1, \ldots, K^{'}$ **do**
15:                 $\pi_{i,r+1} \leftarrow$ solve optimization problem in Eq. 3.
16:                 Store solution $(\pi_{i,r+1}, \boldsymbol{G}^{\pi_{i,r+1}})$ in $\mathcal{X}$.
17:             **end for**
18:         **end for**
19:     **end for**
20:     \\Policy Selection. (Algorithm 1)
21:     $\mathcal{X}_{extension} \leftarrow PolicySelection(N, \mathcal{X})$
22: **end for**
23: \\Policy Assignment.
24: Sample $E$ preferences from $\Omega$.
25: **for** $\boldsymbol{\omega} \in \Omega$ **do**
26:     Solve and find SMP policy via $\pi_{\boldsymbol{\omega}}^{SMP} = \max_{\pi \in \Pi_P} f_{\boldsymbol{\omega}}(\boldsymbol{G}^{\pi})$.
27:     Derive solution $(\pi_{\boldsymbol{\omega}}^{SMP}, \boldsymbol{G}^{\pi_{\boldsymbol{\omega}}^{SMP}})$ for $\boldsymbol{\omega}$.
28: **end for**

---

## B   PROOF OF PROPOSITION 4.3

*Proof.* We prove by contradiction. Suppose that the feasible solution $P^{'} = (\pi', \boldsymbol{G}^{\pi'})$ of problem Eq. 1 is not a Pareto-optimal solution. By the definition of Pareto-optimal solution, there exists a solution $\hat{P} = (\hat{\pi}, \boldsymbol{G}^{\hat{\pi}})$ in $P$ dominates $P'$, i.e., $G_i^{\pi'} \leq G_i^{\hat{\pi}}$ for $\forall i \in [1, 2, \dots, n]$. Assume $P_r = (\pi_r, \boldsymbol{G}_r)$ is the initial point of solving problem Eq. 1. Therefore, $G_l^{\hat{\pi}} \geq G_l^{\pi'} \geq G_l^{\pi_r}$. Because both $P_r$ and $\hat{P}$ and Pareto-optimal solutions, they do not dominate each other. Therefore, given that $G_l^{\hat{\pi}} \geq G_l^{\pi_r}$, there exists $j \in [1, 2, \dots, n], j \neq l$ that $G_j^{\pi_r} \geq G_j^{\hat{\pi}}$.

Now consider the values of $d_j$ and $G_j^{\hat{\pi}}$. Note that $d_j \geq \tilde{G}_j(k-1)$. If $G_j^{\pi_r} \geq G_j^{\hat{\pi}} > d_j$, then $G_j^{\pi_r} \geq G_j^{\hat{\pi}} > \tilde{G}_j(k-1)$, which is conflicting with the condition that $\tilde{G}_j(k-1)$ is the $(k-1)^{th}$ objective value in $\tilde{G}_j$. If $G_j^{\hat{\pi}} \leq d_j$, it conflicts with the assumtion that $\hat{P}$ dominates $P'$. Therefore, $\hat{P}$ does not exist, and $P^{'}$ is a Pareto-optimal solution. $\square$

## C   PROOF OF PROPOSITION 4.4

In this Section, we provide a brief proof for the convergence of iterates majorly inspired by (Tessler et al., 2018). For detailed proof, we refer to Chapter 6 of (Borkar, 2009).

*Proof.* At iteration $r$, the update steps for $\lambda_i$ and $\pi$ are as follows:

$$\lambda_{r+1,i} = \Gamma_\lambda[\lambda_{r,i} - \eta_1(r)\nabla_\lambda L(\lambda_{r,i}, \pi_r)]; \tag{5a}$$
$$\pi_{r+1} = \Gamma_\pi[\pi_r - \eta_2(r)\nabla_\pi L(\boldsymbol{\lambda}_r, \pi_r)]. \tag{5b}$$

Using the log-likelihood trick (Williams, 1992) for the policy update, we have the gradient step

$$\nabla_{\lambda_i} L(\boldsymbol{\lambda}, \pi) = -(d_i - \mathbb{E}_{s\sim\mu}^{\pi_\theta}[G_i^\pi]); \tag{6a}$$
$$\nabla_\pi L(\boldsymbol{\lambda}, \pi) = \nabla_\pi \mathbb{E}_{s\sim\mu}^{\pi_\theta}[\log \pi(s, a; \theta)[R(s) - \boldsymbol{\lambda}^\top \cdot \boldsymbol{G}_{1:n\setminus l}^\pi]]. \tag{6b}$$

In the above, $\eta_1, \eta_2$ are step-sizes which ensure that the policy update is performed on a faster timescale than that of the penalty coefficient $\lambda_i$. We also make the following assumption:

$$\sum_{k=0}^{\infty} \eta_1(k) = \sum_{k=0}^{\infty} \eta_2(k) = \infty, \ \sum_{k=0}^{\infty} (\eta_1(k)^2 + \eta_2(k)^2) < \infty \ \text{ and } \ \frac{\eta_1(k)}{\eta_2(k)} \to 0 \tag{7}$$

Then for the update of policy $\pi$, for any given $\lambda_i$ with the slowest timescale, we can show the following ODE governs the updates:

$$\dot{\pi}_t = \Gamma_\pi(\nabla_\pi L(\boldsymbol{\lambda}, \pi_t)); \tag{8}$$

Similarly, for the update of $\lambda_i$, denote $\pi(\boldsymbol{\lambda}_t)$ as the limiting point of the $\pi$-recursion corresponding to $\lambda_t$, we have the following ODE

$$\dot{\lambda}_{i,t} = \Gamma_{\lambda_i}(\nabla_{\lambda_i} L(\boldsymbol{\lambda}_t, \pi(\boldsymbol{\lambda}_t))). \tag{9}$$

Then following the characterization made by (Tessler et al., 2018) on the internally chain transitive invariant sets of the ODE Eq. 9, we can conclude the convergence for the two-timescale stochastic approximation processes. $\square$

## D   PROOF OF THEOREM 4.5

We reuse the notations for the primal problem Eq. 1 and its dual problem Eq. 2. For the nonconvex primal formulation of MORL, solving its dual problem can only provide an upper bound on $P^*$. Thus it is of interest to evaluate how close the policy obtained by solving the dual is compared to $P^*$. Further, as we resort to interior point method to empirically solve the unconstrained problem, the second part of Proposition 4.5 characterizes the distance between the solution from Eq. 4 and $P^*$.

*Proof.* To show the first part of the Theorem, it majorly relies on the well-established Fenchel-Moreau theorem and the concavity of the perturbed function defined as the following $P(\xi)$:

Denote the duality gap as $\triangle = D^* - P^*$. To show $\triangle = 0$, we first define the perturbation function of problem equation 1:

$$P(\xi) \triangleq \max_{\pi} \ G_l^{\pi}$$
$$\text{s.t.} \quad G_i^{\pi} \geq d_i + \xi_i, \quad i = 1, \ldots, n, \ i \neq l. \tag{10}$$

Then, the conditions under which problem Eq. 1 has zero duality gap are as follows:

**Lemma D.1.** *(Fenchel-Moreau). If (i) Slater's condition holds for (PI) and (ii) its perturbation function $P(\xi)$ is concave, then strong duality holds for (PI).*

In D.1, Slater's condition requires that there exists a feasible policy $\pi$ such that all inequality constraints are strictly satisfied for problem (1), i.e.,

$$G_i^{\pi} > d_i, \quad \forall i = 1, \ldots, n, \ i \neq l.$$

See proof of D.1 in the Corollary. 30.2.2 of (Rockafellar, 1997). For the proof of concavity of the perturbed C-MORL formulation $P(\xi)$, it suffices to show for every $\xi^1, \xi^2 \in \mathbb{R}^n$, and for any $\mu \in (0, 1)$, the following equation holds:

$$P\left[\mu\xi^1 + (1 - \mu)\xi^2\right] \geq \mu P(\xi^1) + (1 - \mu)P(\xi^2). \tag{11}$$

We refer the detailed proof of Eq. 11 in the Proposition 1 of (Paternain et al., 2019).

To show the second part, we make use of the optimality conditions of the dual problem. As the Lagrangian function of Eq. 1 is $L(\boldsymbol{\lambda}, \pi) = G_l^{\pi} - \boldsymbol{\lambda}^{\top}(\mathbf{d} - \boldsymbol{G}_{1:n\backslash l}^{\pi})$, and denote $\tilde{G}_i^{\pi} = d_i - G_i^{\pi}$. Without loss of generality, we follow the standard minimization problem formulation in optimization, and we can write out the dual function as

$$D(\lambda_i) = \min_{\pi} -G_l^{\pi} + \sum_{i=1, i \neq l}^{n} \lambda_i \tilde{G}_i^{\pi} \tag{12}$$

When the problem is strictly feasible, there exists an optimal $\pi^*$ such that for each objective other than the objective $l$, we have $G_i > d_i$. Then the first-order optimality condition holds:

$$-\nabla G_l^{\pi^*} + \sum_{i=1, i \neq l}^{n} \frac{1}{-t \times (\tilde{G}_i^{\pi})} \nabla \tilde{G}_i^{\pi} = 0. \tag{13}$$

We then set $\lambda_i^* = \frac{1}{-t \times (\tilde{G}_i^{\pi})}$ and plug it back into Eq. 13, we can obtain

$$-\nabla G_l^{\pi^*} + \sum_{i=1, i \neq l}^{n} \lambda_i^* \nabla \tilde{G}_i^{\pi} = 0.$$

This shows that under the optimal policy $\pi^*$ and dual variables $\lambda_i^*, i \neq l$, we find the optimal value for the dual function as

$$L(\boldsymbol{\lambda}, \pi^*) = G_l^{\pi^*} - \sum_{i=1, i \neq l}^{n} \lambda_i^* \tilde{G}_i^{\pi} = G_l^{\pi^*} - \frac{n-1}{t}. \tag{14}$$

Using the first-part result that the C-MORL formulation has zero duality gap, $P^* = D(\boldsymbol{\lambda}^*)$, we thus have

$$P^* - G_l^{\pi^*} \leq \frac{n-1}{t}. \tag{15}$$

$\square$

# E EXPERIMENT SETUP DETAILS

## E.1 BENCHMARK

To evaluate the performance of our method and baselines. we collect benchmarks from **MO-Gymnasium** Felten et al. (2023a) and **SustainGym** Yeh et al. (2024). **MO-Gymnasium** is an open source Python library that includes more than 20 environments from diverse domains. **SustainGym** provides standardized benchmarks for sustainable energy systems, encompassing electric vehicles, data centers, electrical markets, power plants, and building energy systems. Among these environments, the building thermal control tasks involve large commercial buildings with multiple zones, aiming to regulate temperature while minimizing energy costs, making them suitable for developing multi-objective control tasks. Additionally, we incorporate the objective of demand response (reducing peak demand) and extend its multi-objective version. The details of all benchmarks are as follows:

**Minecart:** A discrete MORL benchmark with a cart that collects two different and must return them to the base while minimizing fuel consumption. The states of the cart include its $x$ and $y$ position, the current speed, the $sin$ and $cos$ orientations, and the percentage of its occupied capacity by each ore. The agent is allowed to choose between six actions: {*mine, left, right, accelerate, brake, do nothing*}. Let $\not\Vdash$ denote Dirac delta function. The reward of three objectives is defined as:

$$\mathcal{R}_1 = \text{ quantity of ore 1 collected if } s \text{ is inside the base, else } 0;$$
$$\mathcal{R}_2 = \text{ quantity of ore 2 collected if } s \text{ is inside the base, else } 0;$$
$$\mathcal{R}_3 = -0.005 - 0.025 \not\Vdash \{a = \text{ accelerate }\} - 0.05 \not\Vdash \{a = \text{ mine }\}.$$

**MO-Lunar-Lander:** A discrete MORL benchmark with a classic rocket trajectory optimization problem. The state is an eight-dimensional vector that includes the $x$, $y$ coordinates of the lander, its linear velocities in $x$ and $y$, its angle, its angular velocity, and two booleans that represent whether each leg is in contact with the ground or not. The action is a six-dimensional vector: {*do nothing, fire left orientation engine, fire main engine, fire right engine*}. The reward of three objectives is defined as:

$$\mathcal{R}_1 = +100 \text{ if landed successfully}, -100 \text{ if crashed, else } 0;$$
$$\mathcal{R}_2 = \text{ shaping reward };$$
$$\mathcal{R}_3 = \text{ fuel cost of the main engine };$$
$$\mathcal{R}_4 = \text{ fuel cost of the side engines.}$$

**Fruit-Tree:** A discrete MORL benchmark with a full binary tree of depth $d$ with randomly assigned vectorial reward $\mathbf{r} \in \mathbb{R}^6$ on the leaf nodes Yang et al. (2019). These rewards are related to six objectives, showing the amounts of six different nutrition facts of the fruits on the tree:{*Protein, Carbs, Fats, Vitamins, Minerals, Water*}. The goal of the MORL agent is to find a path on the tree from the root to a leaf node that maximizes utility for a given preference. The reward of six objectives is defined as:

$$\mathcal{R}_i = \text{ value of nutrient } i \text{ in } s, \text{ for } i = 1 \dots 6.$$

**MO-Hopper-2d:** The observation and action space are defined as:

$$\mathcal{S} \subseteq \mathbb{R}^{11}, \mathcal{A} \subseteq \mathbb{R}^3.$$

This control task has two conflicting objectives: forward speed and jumping height.

The first objective is forward speed:

$$\mathcal{R}_1 = v_x + C.$$

The second objective is jumping height:

$$\mathcal{R}_2 = 10(h - h_{init}) + C.$$

where $C = -0.001 \sum_i a_i^2$ is composed of extra bonus and energy efficiency, $v_x$ is the speed toward $x$ direction, $h$ is the current height, $h_{init}$ is the initial height, $a_i$ is the action of each actuator.

**MO-Hopper-3d:** The observation and action space are defined as:

$$\mathcal{S} \subseteq \mathbb{R}^{11}, \mathcal{A} \subseteq \mathbb{R}^3.$$

This control task has three conflicting objectives: forward speed, jumping height, and energy efficiency.

The first objective is forward speed:
$$\mathcal{R}_1 = v_x.$$

The second objective is jumping height:
$$\mathcal{R}_2 = 10(h - h_{init}).$$

The third objective is energy efficiency:
$$\mathcal{R}_3 = -\sum_i a_i^2.$$

where $v_x$ is the speed toward $x$ direction, $h$ is the current height, $h_{init}$ is the initial height, $a_i$ is the action of each actuator.

**MO-Ant-2d:** The observation and action space are defined as:
$$\mathcal{S} \subseteq \mathbb{R}^{27}, \mathcal{A} \subseteq \mathbb{R}^8.$$

This control task has two conflicting objectives: x-axis velocity and y-axis velocity.

The first objective is x-axis velocity:
$$\mathcal{R}_1 = v_x.$$

The second objective is y-axis velocity:
$$\mathcal{R}_2 = v_y.$$

where $v_x$ is x-axis speed, $v_y$ is y-axis speed, $a_i$ is the action of each actuator.

**MO-Ant-3d:** The observation and action space are defined as:
$$\mathcal{S} \subseteq \mathbb{R}^{27}, \mathcal{A} \subseteq \mathbb{R}^8.$$

This control task has three conflicting objectives: x-axis speed, y-axis speed, and control cost.

The first objective is x-axis velocity:
$$\mathcal{R}_1 = v_x.$$

The second objective is y-axis velocity:
$$\mathcal{R}_2 = v_y.$$

The third objective is control cost:
$$\mathcal{R}_3 = -2\sum_i a_i^2.$$

where $v_x$ is x-axis speed, $v_y$ is y-axis speed, $a_i$ is the action of each actuator.

**MO-Humanoid-2d:** The observation and action space are defined as:
$$\mathcal{S} \subseteq \mathbb{R}^{376}, \mathcal{A} \subseteq \mathbb{R}^{17}.$$

This control task has two conflicting objectives: forward speed and control cost. MO-Humanoid was chosen because it has one of the most complex state space among all Mujoco environments, with 348 states.

The first objective is forward speed:
$$\mathcal{R}_1 = v_x.$$

The second objective is control cost:
$$\mathcal{R}_2 = -10\sum_i a_i^2.$$

where $v_x$ is the speed in $x$ direction, $a_i$ is the action of each actuator.

**Building-3d:** A building thermal control environment that controls the temperature of a commercial building with 23 zones. The states contain the temperature of multiple zones, the heat acquisition from occupant's activities, the heat gain from the solar irradiance, the ground temperature, the outdoor environment temperature, and the current electricity price. The actions set the controlled heating supply of the zones. The conflicting objectives are minimizing energy cost, reducing the temperature difference between zonal temperatures and user-set points, and managing the ramping rate of power consumption:

$$\mathcal{R}_1 = M - 0.05 * \sum_i |T_i[t] - T_{i,user}[t]|;$$

$$\mathcal{R}_2 = M - 0.05 * \sum_i c[t]|P_i[t]|;$$

$$\mathcal{R}_3 = M - |(\sum_i |P_i[t]| - \sum_i |P_i[t-1]|)|.$$

where $M = 23$ is the number of zones; $T_i$ and $T_{i,user}$ are indoor temperature and user setting point of zone $i$, respectively; $c$ is electricity price; $P_i$ is heating supply power of zone $i$; $t$ is the index of environment time step.

**Building-9d:** A modified version of **Building-3d:**. Instead of calculating the reward based on all zones collectively, this version evaluates the reward for each of the three floors of the commercial building separately. Consequently, this results in a total of $3 \times 3 = 9$ objectives.

Table 6: Environment details of continuous control benchmarks

| Environments | Continuous (Y/N) | Number of Objectives | State Space | Action Space |
|---|---|---|---|---|
| Minecart | N | 3 | $\mathcal{S} \subseteq \mathbb{R}^7$ | $\mathcal{A} \subseteq \mathbb{R}^6$ |
| MO-Lunar-Lander | N | 4 | $\mathcal{S} \subseteq \mathbb{R}^8$ | $\mathcal{A} \subseteq \mathbb{R}^4$ |
| Fruit-Tree | N | 6 | $\mathcal{S} \subseteq \mathbb{R}^2$ | $\mathcal{A} \subseteq \mathbb{R}^2$ |
| MO-Hopper-2d | Y | 2 | $\mathcal{S} \subseteq \mathbb{R}^{11}$ | $\mathcal{A} \subseteq \mathbb{R}^3$ |
| MO-Hopper-3d | Y | 3 | $\mathcal{S} \subseteq \mathbb{R}^{11}$ | $\mathcal{A} \subseteq \mathbb{R}^3$ |
| MO-Ant-2d | Y | 2 | $\mathcal{S} \subseteq \mathbb{R}^{105}$ | $\mathcal{A} \subseteq \mathbb{R}^8$ |
| MO-Ant-3d | Y | 3 | $\mathcal{S} \subseteq \mathbb{R}^{105}$ | $\mathcal{A} \subseteq \mathbb{R}^8$ |
| MO-Humanoid-2d | Y | 2 | $\mathcal{S} \subseteq \mathbb{R}^{348}$ | $\mathcal{A} \subseteq \mathbb{R}^{17}$ |
| Building-3d | Y | 3 | $\mathcal{S} \subseteq \mathbb{R}^{29}$ | $\mathcal{A} \subseteq \mathbb{R}^{23}$ |
| Building-9d | Y | 9 | $\mathcal{S} \subseteq \mathbb{R}^{29}$ | $\mathcal{A} \subseteq \mathbb{R}^{23}$ |

### E.2 TRAINING DETAILS

We run all the experiments on a cloud server including CPU Intel Xeon Processor and GPU Tesla T4. In the Pareto initialization stage, we use PPO algorithm implemented by Kostrikov (2018). The PPO parameters are reported in Table 7 and Table 8. For constrained optimization, we adopt C-MORL-IPO method. For the baseline implementations, **Q-Pensieve** (Hung et al., 2022) and **PG-MORL** (Xu et al., 2020) are reproduced using the official code provided by their respective papers, ensuring consistency with the original experiments. For **Envelope** (Yang et al., 2019), **CAPQL** (Lu et al., 2022), **GPI-LS** (Alegre et al., 2023), and **MORL/D** (Felten et al., 2024), we utilize the implementations available in the MORL-baselines library (Felten et al., 2023a), adapting them as necessary to align with our experimental setup. The hyperparameters of C-MORL-IPO include:

- **Number of initial policy $M$:** the number of initial policies. This parameter is also related to the Pareto initialization stage.
- **Number of extension policy $N$:** the number of policies selected in the Pareto extension stage. This parameter is also related to the Pareto extension stage.
- **Log barrier coef $t$:** the tunable parameter on the log barrier.
- **Constraint relax coef $\beta$:** the constraint relaxing coefficient $\beta$ in Eq. 3.
- **Extension steps $K$:** the number of Pareto extension steps in the Pareto extension stage.
- **Time step:** the total time step contains initialization steps and extension steps. To be specific, $time\_step = timesteps\_per\_actorbatch \times (M + N \times n)$, where $timesteps\_per\_actorbatch$ is a PPO parameter.

The parameters we used are provided in Table 9 and Table 10.

In policy initialization stage, the preference vectors for training initial policies are uniformly sampled from the preference space $\Omega$. For example, in the case of MO-Ant-2d, we set the sampling interval to $0.2$, with the minimum and maximum values for each dimension being $0$ and $1$, respectively. This results in the following preference vectors: $[0, 1], [0.2, 0.8], [0.4, 0.6], [0.6, 0.4], [0.8, 0.2]$, and $[1, 0]$, for a total of 6 preference vectors.

Table 7: PPO Hyperparameters for discrete baselines.

| parameter name | Minecart | MO-Lunar-Lander | Fruit-Tree |
|---|---|---|---|
| steps per actor batch | 512 | 512 | 512 |
| LR($\times 10^{-4}$) | 3 | 3 | 3 |
| LR decay ratio | 1 | 1 | 1 |
| gamma | 0.995 | 0.995 | 0.995 |
| gae lambda | 0.95 | 0.95 | 0.95 |
| num mini batch | 32 | 32 | 32 |
| ppo epoch | 10 | 10 | 10 |
| entropy coef | 0 | 0 | 0 |
| value loss coef | 0.5 | 0.5 | 0.5 |
| max grad norm | 0.5 | 0.5 | 0.5 |
| clip param | 0.2 | 0.2 | 0.2 |

Table 8: PPO Hyperparameters for continuous baselines.

| parameter name | MO-Hopper-2d | MO-Hopper-3d | MO-Ant-2d | MO-Ant-3d | MO-Humanoid-2d | Building-3d | Building-9d |
|---|---|---|---|---|---|---|---|
| steps per actor batch | 512 | 512 | 512 | 512 | 512 | 512 | 512 |
| LR($\times 10^{-4}$) | 3 | 3 | 3 | 3 | 3 | 3 | 3 |
| LR decay ratio | 1 | 1 | 1 | 1 | 1 | 1 | 1 |
| gamma | 0.995 | 0.995 | 0.995 | 0.995 | 0.995 | 0.995 | 0.995 |
| gae lambda | 0.95 | 0.95 | 0.95 | 0.95 | 0.95 | 0.95 | 0.95 |
| num mini batch | 32 | 32 | 32 | 32 | 32 | 32 | 32 |
| ppo epoch | 10 | 10 | 10 | 10 | 10 | 10 | 10 |
| entropy coef | 0 | 0 | 0 | 0 | 0 | 0 | 0 |
| value loss coef | 0.5 | 0.5 | 0.5 | 0.5 | 0.5 | 0.5 | 0.5 |
| max grad norm | 0.5 | 0.5 | 0.5 | 0.5 | 0.5 | 0.5 | 0.5 |
| clip param | 0.2 | 0.2 | 0.2 | 0.2 | 0.2 | 0.2 | 0.2 |

Table 9: C-MORL-IPO Hyperparameters for discrete baselines.

| parameter name | Minecart | MO-Lunar-Lander | Fruit-Tree |
|---|---|---|---|
| M | 6 | 4 | 21 |
| N | 6 | 6 | 6 |
| log barrier coef | 20 | 20 | 20 |
| constraint relax coef | 0.90 | 0.90 | 0.90 |
| Pareto extension step | 60 | 50 | 30 |
| time step ($\times 10^5$) | 5 | 5 | 5 |

### E.3 EVALUATION

We evaluate the quality of Pareto front with the following metrics:

**Definition E.1.** (Hypervolume). Let $P$ be a Pareto front approximation in an n-dimensional objective space and $\boldsymbol{G}_0$ be the reference point. Then the hypervolume metric $\mathcal{H}(P)$ is calculated as $\mathcal{H}(P) = \int_{\mathbb{R}^m} \mathbb{1}_{H(P)}(z)dz$, where $H(P) = \{\boldsymbol{z} \in Z | \exists 1 \leq j \leq |P| : \boldsymbol{G}_0 \preceq \boldsymbol{z} \preceq P(j)\}$. $P(j)$ is the $j^{th}$ solution in $P$, $\preceq$ is the relation operator of objective dominance, and $\mathbb{1}_{H(P)}$ is a Dirac delta function that equals 1 if $z \in H(P)$ and 0 otherwise. A higher hypervolume is better.

**Definition E.2.** (Expected Utility). Let $P$ be a Pareto front approximation in an $n$-dimensional objective space and $\Pi$ be the policy set. The Expected Utility metric is $\mathcal{U}(P) : \mathcal{U}(P) = \mathbb{E}_{\boldsymbol{\omega} \sim \Omega} \left[ \boldsymbol{\omega}^\top \boldsymbol{G}^{\pi_{\boldsymbol{\omega}}^{SMP}} \right]$. A higher expected utility is better.

**Definition E.3.** (Sparsity). Let $P$ be a Pareto front approximation in an $n$-dimensional objective space. Then the Sparsity metric $\mathcal{S}(P)$ is

$$\mathcal{S}(P) = \frac{1}{|P| - 1} \sum_{i=1}^{n} \sum_{k=1}^{|P|-1} \left( \tilde{G}_i(k) - \tilde{G}_i(k+1) \right)^2, \tag{16}$$

where $\tilde{G}_i$ is the sorted list for the $i^{th}$ objective values in $P$, and $\tilde{G}_i(k)$ is the $k^{th}$ value in this sorted list. Lower sparsity is better.

Table 10: C-MORL-IPO Hyperparameters for continuous baselines.

| parameter name | MO-Hopper-2d | MO-Hopper-3d | MO-Ant-2d | MO-Ant-3d | MO-Humanoid-2d | Building-3d | Building-9d |
|---|---|---|---|---|---|---|---|
| M | 6 | 6 | 6 | 6 | 6 | 6 | 9 |
| N | 5 | 6 | 5 | 6 | 5 | 6 | 6 |
| log barrier coef | 20 | 20 | 20 | 20 | 20 | 20 | 20 |
| constraint relax coef | 0.90 | 0.90 | 0.90 | 0.90 | 0.90 | 0.90 | 0.90 |
| Pareto extension step | 60 | 100 | 60 | 100 | 60 | 100 | 100 |
| time step ($\times 10^6$) | 1.5 | 2.0 | 1.5 | 2.0 | 1.5 | 2.0 | 2.5 |

For metrics evaluation, we evenly generate an evaluation preference set in a systematic manner with specified intervals $\Delta = 0.01$, $\Delta = 0.1$, and $\Delta = 0.5$ for benchmarks with two objectives, three or four objectives, and six or nine objectives, respectively. For example, for benchmarks with two objectives, we sample preference vectors covering a range of preferences from $\boldsymbol{\omega} = [0, 1]$ to $\boldsymbol{\omega} = [1, 0]$ with a specified interval $\Delta = 0.01$, totally $1, 01$ preference pairs. For C-MORL and other multi-policy baselines, the calculation of metrics is the same. For single preference-conditioned policy methods, the calculation of metrics is slightly different . Specifically, since in a multi-policy setting, the policies in the Pareto set are preference-irrelevant, we directly use all solutions in the Pareto front to compute hypervolume and sparsity. For single preference-conditioned policy methods, we first evaluate the solutions using all the preferences in the evaluation preference set. Then, the non-dominated solutions that form the Pareto front are used to compute hypervolume and sparsity. Expected utility is evaluated for multi-policy baselines based on the evaluation preference set, and the execution results on all preferences from the evaluation preference set are utilized for single preference-conditioned policy methods.

# F PROCEDURE OF SOLVING C-MORL-CPO AND C-MORL-IPO

## F.1 C-MORL-CPO

**C-MORL-CPO** Rather than applying sampling-based approaches (Duan et al., 2016) for finding policy updates in the relaxed formulation, empirically we can follow the adapted trust region method (Schulman et al., 2015) for CPO updates (Achiam et al., 2017):

$$\pi_{r+1,i} = \arg\max_{\pi \in \Pi_\theta}\{G_l^\pi : G_i^\pi \geq \beta G_i^{\pi_r}, i = 1, \ldots, n, i \neq l; D(\pi, \pi_r) \leq \delta\}. \tag{17}$$

The trust region set is defined as $\{\pi_\theta \in \Pi_\theta : \bar{D}_{KL}(\pi||\pi_r) \leq \delta\}$, where $\bar{D}_{KL}$ denotes the estimated mean of KL-divergence given state $s$, and serves as a surrogate function of the original distance function. Such trust region optimization updates hold guarantees of monotonic performance improvement and constraint satisfaction. It is noteworthy that the original CPO algorithm requires a feasible initialization, which by itself can be very difficult, especially with multiple, general constraints involving policy returns (Zhou et al., 2022). While in our formulation for solving MORL, we can almost guarantee the initial policy is always feasible for the solving process of extended policy with properly selected $\beta$ along with a well-initialized policy set.

However, solving this problem requires evaluation of the constraint functions to determine whether a proposed point $\pi$ is feasible. Therefore, follow (Achiam et al., 2017), we replace the objectives functions with surrogate functions, which are easy to estimate from samples collected on $\pi_r$. To be specific, we solve the following optimization problem to approximately solve Eq. 17:

$$\pi_{r+1} = \arg\max_{\pi \in \Pi_\theta} \mathbb{E}_{s \sim d^{\pi_r}, a \sim \pi}[A^{\pi_r}(s, a)]$$

$$\text{s.t.} \quad G_i^{\pi_r} - \frac{1}{1-\gamma}\mathbb{E}_{s \sim d^{\pi_r}, a \sim \pi}[A_i^{\pi_r}(s, a)] \geq \beta G_i^{\pi_r} \quad i = 1, \ldots, n, i \neq l \tag{18}$$

$$\bar{D}_{KL}(\pi||\pi_r) \leq \delta.$$

However, Eq. 18 is impractical to be solved directly especially when the policy is parameterized as a neural network with high-dimensional parameter spaces. We follow (Achiam et al., 2017) to

implement an approximation to the update Eq. 18 that can be computed efficiently:

$$\boldsymbol{\theta}_{r+1} = \arg\max_{\boldsymbol{\theta}} g^\top(\boldsymbol{\theta} - \boldsymbol{\theta}_r)$$

$$\text{s.t. } c_i + b_i^T(\boldsymbol{\theta} - \boldsymbol{\theta}_r) \le 0 \ i = 1, \dots, n, i \ne l \tag{19}$$

$$\frac{1}{2}(\boldsymbol{\theta} - \boldsymbol{\theta}_r)^T H(\boldsymbol{\theta} - \boldsymbol{\theta}_r) \le \delta.$$

where $g$ is the gradient of the $l^{th}$ objective that is chosen to be optimized, $b_i$ denotes the gradient of the $i^{th}$ objective served as constraint, $H$ is the Hessian of the KL-divergence, and $c_i \doteq \beta G_i^{\pi_r} - G_i^{\pi}$. The dual problem of Eq. 19 can be expressed as:

$$\max_{\substack{\lambda \ge 0 \\ \nu \succ 0}} \frac{-1}{2\lambda}\left(g^T H^{-1}g - 2r\nu + \nu^T S\nu\right) + \nu^T c - \frac{\lambda\delta}{2}, \tag{20}$$

where $B \doteq [b_1, \dots, b_n]$, $c \doteq [c_1, \dots, c_n]^T$, $r \doteq g^T H^{-1}B$, and $S \doteq B^T H^{-1}B$. We derive $\lambda^*$, $\nu^*$ by solving the dual, then the solution to the primal Eq. equation 19 is

$$\boldsymbol{\theta}^* = \boldsymbol{\theta}_r + \frac{1}{\lambda^*}H^{-1}(g - B\nu^*). \tag{21}$$

In our implementation, we adopt C-MORL-CPO to solve MORL tasks with two objectives, i.e., constrained optimization problem *that only involves single constraint*. Therefore, we can directly compute the dual variables $\lambda^*$ and $\nu^*$ with the analytical solution as follows Achiam et al. (2017):

$$\nu^* = \left(\frac{\lambda^* c - r}{s}\right)_+,$$

$$\lambda^* = \arg\max_{\lambda \ge 0} \begin{cases} f_a(\lambda) \doteq \frac{1}{2\lambda}\left(\frac{r^2}{s} - q\right) + \frac{\lambda}{2}\left(\frac{n^2}{s} - \delta\right) - \frac{rr}{s} & \text{if } \lambda c - r > 0 \\ f_b(\lambda) \doteq -\frac{1}{2}\left(\frac{q}{\lambda} + \lambda\delta\right) & \text{otherwise,} \end{cases} \tag{22}$$

with $q = g^T H^{-1}g$, $r = g^T H^{-1}b$, and $s = b^T H^{-1}b$. If the constrains in Eq.18 are not satisfied, the constrained optimization towards the corresponding objective direction in the current Pareto extension iteration will be terminated.

### F.2   C-MORL-IPO

Recall in Section 5.3, we augment the objective function of the objective that is being optimized with logarithmic barrier functions for other constrained objectives. Note that the logarithmic barrier functions can be integrated with any other policy optimization methods, in this paper, we follow (Liu et al., 2020) to integrate it with PPO (Schulman et al., 2017) for training our policies. Therefore, the surrogate objective becomes:

$$\max \ L^{CLIP}(\theta) + \sum_{i=1}^{n}\phi(G_i^\pi) \tag{23}$$

where $\phi(G_i^\pi) = \frac{\log(G_i^\pi - \beta G_i^{\pi_r})}{t}$ and $t$ is a hyperparameter; $L^{CLIP}(\theta)$ is the clipped surrogate objective of PPO. As $t$ approaches $\infty$, the approximation becomes closer to the indicator function.

## G   ADDITIONAL RESULTS

### G.1   PARAMETER STUDY FOR NUMBER OF EXTENSION POLICIES

Table 11: Ablation study of C-MORL for the number of extension policies on MO-Ant-3d benchmark.

|  | N=6 | N=12 | N=18 |
|---|---|---|---|
| **HV**$_{(10^7)}$ | 4.03±0.17 | **4.21±0.22** | 4.20±0.27 |
| **EU**$_{(10^2)}$ | 2.59±0.08 | **2.65±0.08** | 2.63±0.11 |
| **SP**$_{(10)}$ | 2.91±0.90 | 2.00±0.27 | **1.53±0.23** |

It is more difficult to derive Pareto-optimal policies to cover the entire Pareto front for the continuous MORL tasks with more than two objectives. In this subsection we look into different settings for

C-MORL and associated impacts to the algorithm performances. Table 11 presents the results of the ablation study for the number of extension policies on the MO-Ant-3d benchmark, which is one of the continuous MORL tasks with numerous objectives. We can observe that when $N = 12$, C-MORL can derive a Pareto front with much higher hypervolume and lower sparsity than that of $N = 6$. When $N = 18$, the result is similar, indicating that 12 policies are sufficient to be extended to fill the Pareto front for this task. Fig. 7 further illustrates the Pareto extension process with varying values of the number of extension policies ($N = 6, 12, 18$) on the MO-Ant-3d benchmark. It can be observed that as the number of policies selected to be extended increases, the hypervolume also increases. Notably, Fig. 7 highlights that when $N = 18$, there is a more comprehensive exploration of the objectives of Y velocity. This observation suggests a more thorough exploration of sparser areas on the Pareto front.

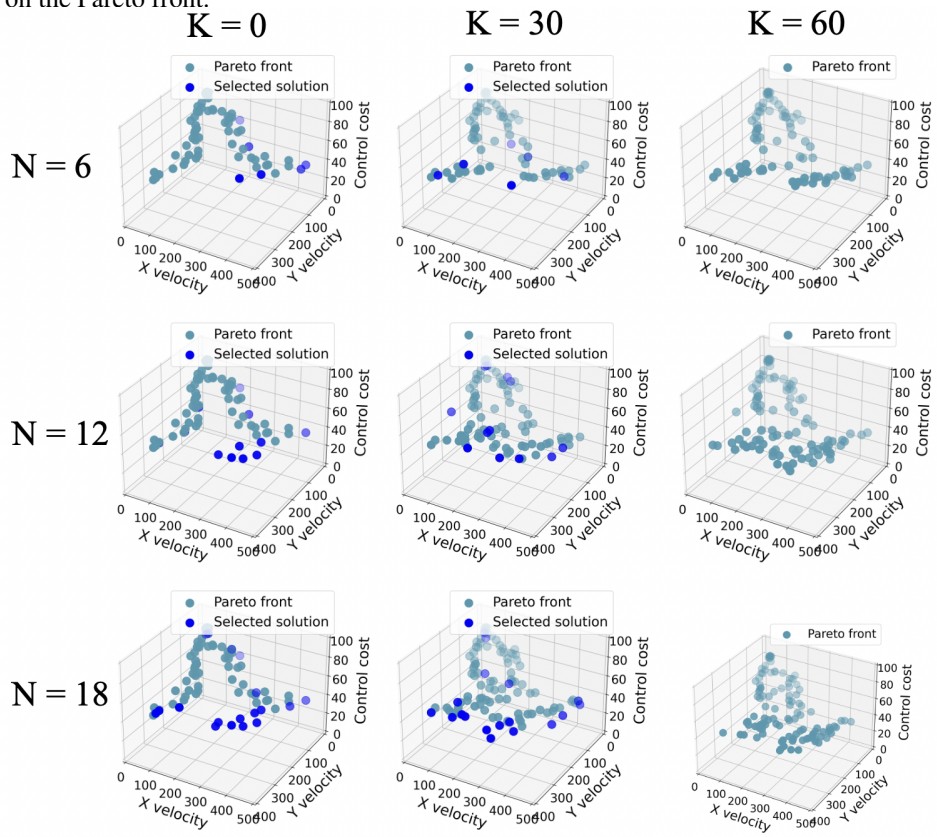

Figure 7: Ablation study of C-MORL for the number of extension policies on MO-Hopper-v3 benchmark. Number of extension policies on MO-Ant-3d benchmark $N = 6, 12, 18$ and number of Pareto extension step $K = 0, 30, 60$, respectively.

### G.2 ABLATION STUDY ON BUFFER SIZE.

In order to better understand how Buffer size can influence the performance of C-MORL, we provide the comparison of the best baseline and C-MORL with various buffer sizes, as shown in Table 12 and Table 13. These studies show that even with reduced buffer sizes, C-MORL maintains highly competitive performance. Compared to PG-MORL, our C-MORL achieves superior results while requiring significantly fewer policies. Another notable observation is that, in some benchmarks, the results remain consistent regardless of the buffer size. This consistency indicates that the number of Pareto optimal policies in these cases does not exceed the buffer size.

### G.3 EXPECTED UTILITY RESULTS

C-MORL outperforms other methods in terms of expected utility across nine out of ten benchmarks, which can be attributed to two significant advantages. The first advantage is related to the inherent characteristics of the multi-policy approach. As illustrated in Fig. 8, single preference-conditioned

Table 12: Evaluation of HV, EU, and SP in discrete MORL tasks under varying buffer sizes, alongside a comparison with the best-performing baseline.

| Environments | Metrics | Best Baseline | B=20 | B=50 | B=100 | B=200 |
|---|---|---|---|---|---|---|
| Minecart | $HV(10^2)$ | **6.05±0.37(GPI-LS)** | 6.22±0.70 | **6.57±0.92** | **6.63±0.89** | **6.77±0.88** |
| | $EU(10^{-1})$ | **2.29±0.32(GPI-LS)** | 1.88±0.59 | **2.05±0.67** | **2.09±0.65** | **2.12±0.66** |
| | $SP(10^{-1})$ | 0.10±0.00(GPI-LS) | 0.64±0.17 | 0.19±0.02 | 0.09±0.03 | **0.05±0.02** |
| MO-Lunar-Lander | $HV(10^9)$ | 1.06±0.16(GPI-LS) | 1.01±0.07 | **1.08±0.04** | **1.12±0.03** | **1.12±0.03** |
| | $EU(10^1)$ | 1.81±0.34(GPI-LS) | 1.84±0.31 | **2.21±0.23** | **2.35±0.18** | **2.35±0.18** |
| | $SP(10^3)$ | **0.13±0.01(GPI-LS)** | 1.14±2.14 | 1.65±1.83 | 1.04±1.24 | 1.04±1.24 |
| Fruit-Tree | $HV(10^4)$ | **3.66±0.23(Envelope)** | 2.34±0.29 | 2.86±0.19 | 3.17±0.20 | **3.67±0.14** |
| | EU | 6.15±0.00(Envelope/GPI-LS) | 6.14±0.13 | 6.38±0.10 | 6.46±0.08 | **6.53±0.03** |
| | SP | 0.53±0.02(GPI-LS) | 2.62±0.49 | 1.65±1.83 | 0.81±0.12 | **0.04±0.00** |

Table 13: Evaluation of HV, EU, and SP in continuous MORL tasks under varying buffer sizes, alongside a comparison with the best-performing baseline.

| Environments | Metrics | Best Baseline | B=20 | B=50 | B=100 | B=200 |
|---|---|---|---|---|---|---|
| MO-Hopper-2d | $HV(10^5)$ | 1.26±0.01(Q-Pensieve) | **1.39±0.01** | **1.39±0.01** | **1.39±0.01** | - |
| | $EU(10^2)$ | 2.34±0.10(PG-MORL) | **2.55±0.01** | **2.56±0.02** | **2.56±0.02** | - |
| | $SP(10^2)$ | **0.46±0.10(CAPQL)** | 2.68±1.66 | 0.57±0.29 | 0.33±0.28 | - |
| MO-Hopper-3d | $HV(10^7)$ | 1.70±0.29(GPI-LS) | 2.03±0.15 | 2.20±0.04 | 2.26±0.02 | **2.29±0.01** |
| | $EU(10^2)$ | 1.62±0.10(GPI-LS) | 1.72±0.08 | 1.78±0.02 | **1.80±0.01** | **1.80±0.01** |
| | $SP(10^2)$ | 0.74±1.22(GPI-LS) | 7.61±2.24 | 2.74±1.91 | 0.86±0.27 | **0.28±0.09** |
| MO-Ant-2d | $HV(10^5)$ | **3.10±0.25(GPI-LS)** | 3.08±0.21 | **3.13±0.20** | **3.13±0.20** | - |
| | $EU(10^2)$ | **4.28±0.19(GPI-LS)** | 4.27±0.19 | **4.29±0.19** | **4.29±0.19** | - |
| | $SP(10^3)$ | **0.18±0.07(CAPQL)** | 3.66±1.29 | 1.67±0.85 | 1.67±0.85 | - |
| MO-Ant-3d | $HV(10^7)$ | 3.82±0.43(Q-Pensieve) | 3.52±0.16 | 3.83±0.17 | **4.00±0.12** | **4.09±0.13** |
| | $EU(10^2)$ | 2.41±0.20(GPI-LS) | 2.48±0.09 | **2.55±0.08** | **2.57±0.07** | **2.57±0.06** |
| | $SP(10^3)$ | **0.02±0.01(PG-MORL)** | 1.56±0.67 | 0.30±0.19 | 0.08±0.05 | **0.03±0.01** |
| MO-Humanoid-2d | $HV(10^5)$ | 3.30±0.05(CAPQL) | **3.43±0.06** | **3.43±0.06** | **3.43±0.06** | - |
| | $EU(10^2)$ | **4.75±0.04(CAPQL)** | **4.78±0.05** | **4.78±0.05** | **4.78±0.05** | - |
| | $SP(10^4)$ | 0±0.00(CAPQL/GPI-LS) | 0.18±0.27 | 0.18±0.27 | 0.18±0.27 | - |
| Building-3d | $HV(10^{12})$ | 1.00±0.02(Q-Pensieve) | 1.14±0.00 | 1.14±0.00 | **1.15±0.00** | **1.15±0.00** |
| | $EU(10^4)$ | 0.96±0.00(Q-Pensieve) | **1.02±0.00** | **1.02±0.00** | **1.02±0.00** | **1.02±0.00** |
| | $SP(10^4)$ | **0.37±0.22(PG-MORL)** | 1.04±0.16 | 1.98±0.38 | 0.69±0.62 | 0.69±0.62 |
| Building-9d | $HV(10^{31})$ | 7.28±0.57(Q-Pensieve) | 7.64±0.17 | **7.93±0.07** | **7.93±0.07** | **7.93±0.07** |
| | $EU(10^3)$ | 3.46±0.03(Q-Pensieve) | 3.50±0.00 | **3.52±0.00** | **3.52±0.00** | **3.52±0.00** |
| | $SP(10^4)$ | **0.10±0.04(Q-Pensieve)** | 1.16±0.19 | 0.28±0.03 | 0.28±0.04 | 0.28±0.04 |

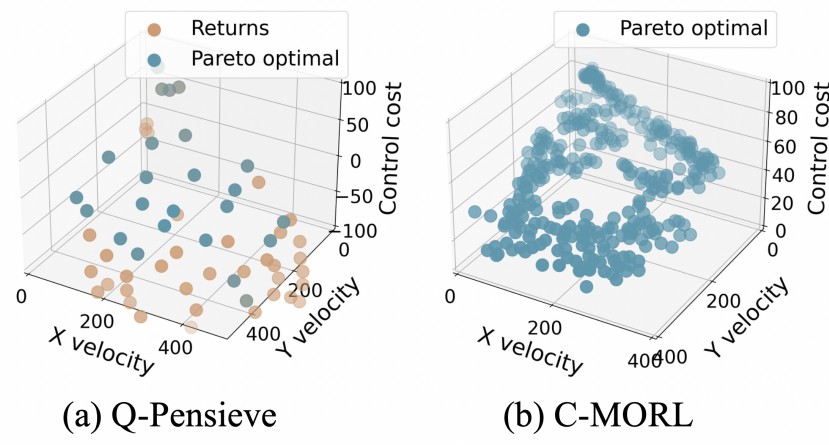

(a) Q-Pensieve        (b) C-MORL

Figure 8: Evaluation results of sampled preferences on MO-Ant-3d benchmark. (a) Q-Pensieve evaluation. Returns of evaluated preference pairs with interval 0.1 are marked with orange points, while Pareto-optimal solutions are marked with blue points. (b) C-MORL evaluation. Pareto-optimal solutions are marked with blue points.

policy approach does not guarantee that each preference sampled to be evaluated corresponds to a solution on the Pareto front. The Pareto front exclusively encompasses solutions that are Pareto optimal, while the majority of preferences do not lead to such optimal solutions. This limitation arises from the inherent difficulty of achieving Pareto optimal for every preference in this kind of approach. In contrast, the multi-policy approach yields a Pareto solution set that exclusively comprises a Pareto-optimal solution set that is irrelevant of specific preferences. Consequently, when presented with an unseen preference, one can simply select the solution with the utility (i.e. the SMP in Eq. 3.1) from this pre-existing set.

## H    Discussion and Limitations of C-MORL

With the novel design of Pareto initialization, policy selection, and Pareto extension, the proposed method gives a novel and systematic approach on exploring the Pareto front and optimizing the policies. Despite the capability of our approach to effectively populate the Pareto front, we observe that in some benchmarks, the current constrained policy optimization method fails to adequately extend the policies toward certain objective directions. Consequently, there still exist unexplored areas on the Pareto front. To address this issue, we plan to develop a more effective extension method. Additionally, since our implementation is based on PPO, its training efficiency is outstanding, but its sample efficiency is relatively low. In the future, we plan to develop more sample-efficient methods for MORL.

