# OpenReview forum: "Efficient Discovery of Pareto Front for Multi-Objective Reinforcement Learning"
_ICLR.cc/2025/Conference — ICLR 2025 Poster_

### Official Review · Reviewer_EBHH · 2024-10-23

**Soundness:** 2
**Presentation:** 1
**Contribution:** 2
**Rating:** 5
**Confidence:** 4

**Summary:**

This paper introduces C-MORL, a two-stage policy optimization algorithm aimed at efficiently discovering the Pareto front through a constrained optimization framework for multi-objective reinforcement learning (MORL). C-MORL effectively manages complex tasks with multiple discrete and continuous objectives, as demonstrated in experiments involving up to nine objectives. To achieve solutions without additional computational costs, the paper presents an interior-point-based approach for solving a relaxed formulation, ensuring the derivation of Pareto-optimal policies under specified conditions. Extensive evaluations across diverse MORL benchmarks, including robotics control and sustainable energy management, highlight C-MORL’s superior capability to explore a broader Pareto front under unseen preferences.

**Strengths:**

The paper builds a solid foundation for its optimization approach by employing extensive constraint theories. The experimental section thoroughly evaluates state-of-the-art MORL methods on multi-objective optimization problems, with a comprehensive analysis specifically on the newly introduced Building-9D benchmark. The proposed two-stage method partially addresses certain challenges in multi-objective optimization (MOO). The conclusions align well with the research questions posed, demonstrating consistency between the methodology and findings.

**Weaknesses:**

1. **Lack of Innovation and Convincing Results:**
   The structure of the paper is reasonable, but the main idea and innovation seem insufficient. The contribution appears to be limited to encapsulating PGMORL or any single RL algorithm by using crowding distance to select policies for constrained optimization. For preferences that have not been trained, the method merely matches them to an existing best policy without discovering new policies corresponding to those preferences. This results in policies that are locally optimal rather than truly globally optimal in the CCS, which is unconvincing.

2. **Selection of Extreme Solutions:**
   Since extreme solutions have already reached performance bottlenecks on particular objectives, selecting them by default does not facilitate the exploration of the potential Convex Coverage Set (CCS). This approach may limit the discovery of more diverse and optimal policies.

3. **Efficiency and Resource Intensity:**
   The main idea—selecting policies to train based on crowding distance and storing all intermediate policies in a buffer—is straightforward. However, constructing the CCS through a vast number of policies is highly resource-intensive and inefficient. This method may not be practical for large-scale applications due to its computational demands.

4. **Time Complexity Comparison:**
   Given that the work uses PGMORL for initialization, comparing the time complexity to PGMORL is not entirely fair since it is already part of the proposed algorithm. A more appropriate comparison would isolate the additional computational overhead introduced by the proposed method.

5. **Unclear Initialization and Preference Distribution:**
   The paper does not clearly explain how many policies are selected during initialization, how many corresponding weight preferences are chosen, and how they are distributed—an important aspect of the methodology. In subsequent policy selection and optimization, the distribution of preferences remains unchanged. This could lead to policies converging around certain preferences, hindering further exploration of the CCS. While PGMORL addresses this issue effectively with a utility prediction method, your paper does not mention the problem. Consequently, it is unclear whether the filling of CCS gaps in the experimental results belongs to the same cluster.

6. **Overuse of Formulas and Theorems:**
   The paper includes a large number of formulas and theorems, which detracts from the reading experience. These mathematical details contribute little to the core algorithm, especially regarding policy selection and assignment. Simplifying or reducing the emphasis on these formulas could improve clarity and accessibility.

7. **Inconsistency in Method Presentation:**
   Similar to the previous point, C-MORL-IPO and C-MORL-CPO are defined with extensive formulas and theorems, but only C-MORL is used in the experimental results. It is unclear which method is employed in the policy extension process of C-MORL (i assume IPO?). These two methods do not appear in the baselines, nor are there corresponding ablation studies to support them, i.e., there is no evidence in the main results support the claim 'C-MORL-IPO is more computationally efficient than C-MORL-CPO'. Their reappearance in the supplementary materials without integration into the main results creates a fragmented reading experience and causes confusion.

8. **Statistical Significance in Comparison Results:**
   In the comparison results table for the MO-Ant-2d environment under HV and EU metrics, the error of the standard deviation between C-MORL and GPI-LS exceeds the error of mean difference. Therefore, it cannot be considered as a solid surpassing to the baseline.

**Questions:**

1. **Selection of Mujoco Environments:**
   Why were only certain environments from the Mujoco suite selected as baselines for demonstration while other significant environments were omitted?

2. **Unclear Initialization and Preference Distribution:**
   The paper does not clearly explain how many policies are selected during initialization, how many corresponding weight preferences are chosen? And how they are distributed?

3. **Efficiency and Resource Intensity:**
The main idea of selecting policies for training based on crowding distance and storing all intermediate policies in a buffer is straightforward. However, constructing the CCS through a vast number of policies seems highly resource-intensive and inefficient. In each MO-problem, How many policies do you store, and how many are used to construct the CCS? Compared to PGMORL, do you utilize more or fewer policies in this process?

4. **Selection of Extreme Solutions:**
   Since extreme solutions have already reached performance bottlenecks on particular objectives, selecting them by default does not facilitate the exploration of the potential Convex Coverage Set (CCS). Why are they selected as default for next training round?

---

> ### Author Response · Authors · 2024-11-23
>
> We sincerely appreciate your thoughtful questions. Below, we first clarify a few interpretations and claims shown in this paper, and then provide further discussions:
>
> **(D-1) Lack of Innovation and Convincing Results.**
>
> **Novelty and advantages.**
>
> We want to justify this work’s novelty from the following two perspectives.
>
> **1. Framework.** The key innovation of our algorithm lies in its ability to efficiently construct the full Pareto front by constrained policy optimization-based Pareto extension starting from only a small number of points on it. These initial points are obtained by training a limited set of policies to derive Pareto optimal solutions. This also allows us to bridge rich tools and theories from constrained optimization to MORL problems. C-MORL is not encapsulating PG-MORL, as C-MORL distinguishes itself from Pareto extension, constrained optimization formulation and solution approaches, as well as superior efficiency (detailed below). And as mentioned in Section 2. Related Work, this also holds a fundamental distinction from single preference-conditioned policy or multi-policy approaches in the literature.
>
> **2. Training efficiency.** From only a small number of Pareto optimal policies, C-MORL employs a constrained policy optimization-based Pareto extension to systematically derive new policies, completing the Pareto front without the need to train additional policies from scratch. This approach significantly reduces computational overhead while ensuring high-quality solutions across the entire Pareto front. We also show a novel 9-objective environment, where PG-MORL fails short of finding good policies.
>
> **3. Ability to extend to benchmarks with large objective spaces.** Existing methods primarily focus on tasks with continuous observation spaces and up to 5 objectives, or discrete observation spaces with up to 6 objectives. In contrast, C-MORL can effectively handle benchmarks with larger objective spaces by identifying gaps on the Pareto front and generating new policies in the desired regions. This is achieved by optimizing a policy toward a specific objective direction while constraining the performance of other objectives, enabling precise and efficient expansion of the Pareto front.
>
> **Comparison with PG-MORL.** The most significant limitation of PG-MORL [1] is its inability for benchmarks with more objectives, as the computation time grows exponentially when number objectives increase. As a typical evolutionary approach, in each iteration, it involves solving a knapsack problem using a greedy algorithm to identify tasks that could most effectively improve Pareto quality. This step has an exponential time complexity as the number of objectives increases. Instead of iteratively improving the overall quality of a policy set as most evolutionary approaches do, C-MORL directly fills the desired region of Pareto front via constrained policy optimization from a few Pareto optimal policies.
>
> **Convex-Converage-Set (CCS).** By definition, a Convex Coverage Set (CCS) is a finite convex subset of the Pareto front, where each policy in the set is optimal with respect to some linear utility function [2]. This means that for any linear preference, there exists a policy within the CCS that is optimal. It is important to note that the CCS is a subset of the Pareto front, and the term "optimal" in its definition refers only to optimality within the CCS, not the entire Pareto set.
>
> In most benchmarks, particularly those with continuous objectives, obtaining the complete Pareto set is intractable. This limitation highlights the need to select the best policy from the set for a given preference. Ideally, increasing the buffer size could improve the quality of the Pareto set, resulting in higher expected utility for specific preferences. However, this approach introduces a trade-off between efficiency and performance.

---

> ### Author Response · Authors · 2024-11-23
>
> **(D-2) Time complexity comparison.** We would like to clarify that PG-MORL is not part of our algorithm, and as such, PG-MORL’s computation time should not be included in the analysis. A key and time-consuming component of PG-MORL is their prediction-guided policy selection, which involves solving a knapsack problem using a greedy algorithm to identify tasks that could most effectively improve Pareto quality. This step has an exponential time complexity as the number of objectives increases. Also as shown in Table 2, PG-MORL is time out for the Building-9d case.
>
> In contrast, proposed C-MORL does not need to solve this knapsack problem; instead, it identifies the gaps on Pareto front based on the crowding distance of policies. Moreover, the overall procedure of C-MORL is fundamentally different from PG-MORL. While PG-MORL begins with a brief initialization phase, followed by an evolutionary process involving policy selection and training, C-MORL takes a different approach. It trains a small number of policies until near convergence and then performs constrained policy optimization to fill Pareto front.
>
> As a result, the major computation of C-MORL primarily comes from solving multiple constrained policy optimization problems, as analyzed in Proposition 5.1.
>
> **(D-3) Policies converging around certain preferences.** The reviewer mentions that ‘In subsequent policy selection and optimization, the distribution of preferences remains unchanged’. We would like to clarify that the **policy selection and optimization process do not rely on any specific preference**. Instead, based on the crowd distance, we identify the sparse areas of the Pareto front, and then conduct constrained policy optimization to fill these gaps.
>
> Next, we answer other questions and weaknesses raised by the reviewer.
>
> **(D-4) (Weakness 7) Inconsistency in Method Presentation.** As stated in Appendix E.2, , we implement all of our experiments for C-MORL based on C-MORL-IPO. This is because IPO is a first-order method, which is more computationally efficient than C-MORL-CPO. We mentioned both algorithms since they can both give tractable solutions for the MORL formulation. And these two methods’ major differences are on the solving procedure of optimization rather than on the solution quality. In the revised manuscript, we made further explanations on the experiment settings, and reorganized the paper writings to make the algorithm part easier to follow.
>
> **(D-5) (Weakness 8) Statistical Significance in Comparison Results.** Thank you for pointing this out; we will revise them accordingly. In our revised version, a method is also marked if its mean value lies within the confidence interval of the best method. We would like to note that C-MORL’s performance is still most consistent across all metrics and environments as shown in Table 1 and Table 2.
>
> **(D-6) (Question 1) Selection of Mujoco Environments.** In the manuscript, we selected environments from the widely used and comprehensive MORL toolkit, mo-gymnasium, prioritizing the most challenging baselines as documented in the mo-gymnasium reference materials.
>
> Specifically:
>
> **MO-Humanoid** was chosen because it has the most complex state space among all Mujoco environments, with 348 states, making it very challenging for many MORL algorithms.
>
> **MO-Ant** and **MO-Hopper** were selected for their diverse number of objectives (2 and 3, respectively) and relatively complex state spaces. For instance, MO-Ant has 105 states, which are significantly more than those in simpler environments we excluded.
>
> For environments we did not select, such as **MO-Swimmer**, which has only 8 states and 2 objectives, we believe they are relatively easy to train and thus do not present sufficient challenge for meaningful evaluation.
>
> **(D-7) (Question 2) Unclear Initialization and Preference Distribution.** We are sorry for the confusion made in the manuscript. In Appendix E.2, we discussed training details, including the number of initial policies $M$, as shown in Table 8 and 9. In the policy initialization stage, the preference vectors for training initial policies are uniformly sampled from the preference space $\Omega$. For example, in the case of MO-Ant-2d, we set the sampling interval to 0.2, with the minimum and maximum values for each dimension being 0 and 1, respectively. This results in the following preference vectors: [0, 1], [0.2, 0.8], [0.4, 0.6], [0.6, 0.4], [0.8, 0.2], and [1, 0], for a total of 6 preference vectors. In the revised manuscript, we added details on the preference vectors.

---

> ### Author Response · Authors · 2024-11-23
>
> **(D-8) (Question 3) Efficiency and Resource Intensity.** We would like to clarify that C-MORL did not store all intermediate policies. Instead, we maintained a policy buffer. If the number of policies exceeds the policy buffer size in the current iteration, we select a number of policies equal to buffer size from the set of all Pareto optimal policies based on their crowd distance.
>
> For Table 1 and 2, we set the buffer size $B=100$ for tasks with 2 objectives and $B=200$ for tasks with 3 or more objectives, following the same configuration as PG-MORL. However, we observed that in many benchmarks, the number of policies never exceeded buffer size and was significantly smaller. This suggests that C-MORL is not resource-intensive.
>
> To further analyze the influence of buffer size and also demonstrate the resource efficiency of C-MORL, we conducted additional ablation studies on buffer size, which are detailed in Table 12 and Table 13 in ‘Official comment by Authors’ for all reviewers.  These studies show that even with reduced buffer sizes, C-MORL maintains highly competitive performance.
>
> Compared to PG-MORL, our C-MORL achieves superior results while requiring significantly fewer policies. Another notable observation is that, in some benchmarks, the results remain consistent regardless of the buffer size. This consistency indicates that the number of Pareto optimal policies in these cases does not exceed the buffer size.
>
> **(D-9) (Question 4) Selection of Extreme Solutions.** This is because for fair comparison, we ran C-MORL and baselines with the same number of steps (as mentioned in Experiment setup, $5 \times 10^5$ time steps for discrete benchmarks. Continuous benchmarks with two, three, and nine objectives are trained for $1.5 \times 10^6$, $2 \times 10^6$, $2.5 \times 10^6$ steps, respectively. Therefore, although ideally we would like to derive some Pareto optimal solutions in the Pareto initialization stage, while training steps are limited, the initial policies have not reached bottlenecks in some of the benchmarks in the initialization stage. We empirically find that if these policies are selected (only in the first extension iteration), the overall results can be slightly better.
>
> [1] Xu, Jie, et al. "Prediction-guided multi-objective reinforcement learning for continuous robot control." International conference on machine learning. PMLR, 2020.
>
> [2] Alegre, Lucas Nunes, Ana Bazzan, and Bruno C. Da Silva. "Optimistic linear support and successor features as a basis for optimal policy transfer." International conference on machine learning. PMLR, 2022.

---

> > ### Author Response · Authors · 2024-11-25
> >
> > Thank you for your feedback and for reviewing our work. We want to kindly check if there are any additional concerns or comments from your side. Please feel free to let us know if we can provide further clarification. Thank you again for your time and efforts.

---

> > > ### Comment · Reviewer_EBHH · 2024-11-27
> > >
> > > Thanks for the detailed response. I have adjusted the score accordingly. Good luck

---

> > > > ### Author Response · Authors · 2024-11-28
> > > >
> > > > Thank you for recognizing the improvements in our paper. We greatly appreciate your constructive feedback and support.

---

### Official Review · Reviewer_dWup · 2024-10-26

**Soundness:** 1
**Presentation:** 2
**Contribution:** 2
**Rating:** 3
**Confidence:** 4

**Summary:**

This paper introduces a novel method for learning a Pareto set of policies for solving multi-objective reinforcement learning (MORL) problems in which the reward is a vector composed of multiple conflicting objectives. The paper introduces Constrained MORL (C-MORL), an algorithm that maintains a population of policies to approximate the Pareto front. At each iteration, it learns novel policies that are optimized to solve a constrained MDP such that the solution to it lies in the Pareto front. The method was evaluated in several benchmarks with varying numbers of objectives and characteristics in terms of different MORL evaluation metrics.

**Strengths:**

- The problem of deciding which directions in the objective space are more relevant to learn when approximating a Pareto front is very relevant to the MORL community.
- The evaluation considers many challenging benchmarks with discrete and continuous action spaces and up to 9 objectives.

**Weaknesses:**

- The paper can significantly be improved in terms of clarity (see below).
- The method requires learning a separate neural network for each policy, which does not scale in terms of memory. This is in contrast with previous MORL algorithms that train a single model conditioned on the weight vector $w$.
- There are many experimental details missing in the paper that need to be addressed (see Questions below).

**Questions:**

Below, I have questions and constructive feedback to the authors:

1) “(Alegre et al., 2023) further introduce a novel Dyna-style MORL method that significantly improves sample efficiency while relying on accurate learning of dynamics”.
They also introduce a model-free version of their algorithm (GPI-LS), which parallelizes the training of the CCS.

2) Minor comment about the notation: The letter $G$ is usually used in the RL literature to denote the random variable of the return, and not its expected value. The expected value is usually denoted with the letter $J$ or by $V(S_0)$. I would suggest considering changing this notation to avoid confusion.

3) In the definition of the space of preference weights, $\Omega$, you should also restrict $w_i\geq0$.

4) It is not clear how the projection operators $\Gamma_\lambda$ and $\Gamma_\pi$ are defined. What does it mean to project $\lambda$ and $\pi$ into a compact and convex set?

5) In Proposition 4.4, it is not clear how the policy update works. Are you assuming $pi_r$ is a parameterized policy? Is this equation updating the parameters of the policy? What is a “fixed point of MORL policy”?

6) In Theorem 4.5, what is the Slater’s condition?

7) “we also maintain a solution buffer to enhance policy diversity.” What is a solution buffer, and how does it enhance policy diversity? Tihs is not explained in this section.

8) In Eq. 3, how is the hyperparameter $\beta$ selected? How can it impact the optimality of the method, in theory?

9) In Eq. 5, how is the parameter $t$ of the function $\phi$ selected?

10) I am not sure Proposition 5.1 is making a fair statement. Although the algorithm running time is linear in the number of extension policies $N$, it is expected that the number of policies $N$ required to cover the Pareto front would probably increase exponentially with the number of objectives. Of course, a user can use the same value of $N$ independently of the number of objectives $n$, but that would probably be sub-optimal. That is, it is expected that one should increase $N$ as $n$ increases.

11) It is not very clear in the paper when C-MORL-CPO or C-MORL-IPO are used. Which one is used in the proposed algorithm?

12) Why does PG-MORL have to solve a knapsack problem with $K$x$N$ points? This is not mentioned in the paper of Xu et al., 2020.

13) In Table 2, the authors denote in bold some metrics which are overlapping with the competitors. For instance, in MO-Ant-2d, HV of $3.10\pm0.25$ and $3.13\pm0.20$ and EU $4.28\pm0.19$ and $4.29\pm0.19$ overlap.

14) There are many experimental details missing in the experimental section:
- How many random seeds were used in each experiment? Are the metrics averaged over how many runs?
- Which version of CAPQL and GPI-LS did the authors use? How were the hyperparameters of the competitors chosen?
- CAPQL and GPI-LS are based on SAC and TD3, while C-MORL is based on PPO. This could significantly impact the results, and it makes it difficult to infer whether the performance gains come from SAC vs. PPO or the expansion technique introduced in the paper.
- Alternatively, did the authors reimplement CAPQL and GPI-LS to use PPO as the underlying policy learning technique?
- How were the PFs in Figure 4 generated? Are those points average over how many episodes? How many weight vectors were used to evaluate the learned policies for different utilities?

15) The authors mention that “each method is tested across three
runs with the same seeds”. Three runs are very unlikely to be sufficient to provide significant statistical results.

16) I suggest including the ablation of the Pareto Extension procedure and comparing it with solving for random linear weights since this procedure is arguably more important than the Policy Selection procedure. It would be important to observe if the overhead of employing CPO or IPO is really providing performance gains.

17) The idea of the paper of solving constrained MDPs to identify new policies in the Pareto front is similar to that of (Willem et al., 2024, “Divide and Conquer: Provably Unveiling the Pareto Front with Multi-Objective Reinforcement Learning”). Please add a discussion explaining the similarities and differences.

**Details Of Ethics Concerns:**

I identified no ethical concerns that need to be addressed in this paper.

---

> ### Author Response · Authors · 2024-11-23
>
> We appreciate your constructive feedback.
>
> **(C-1) The training of GPI-LS is definitely not parallel.** In this sentence we would like to clarify the sample efficiency of GPI-PD. Meanwhile, its model-free version, GPI-LS,  never parallelizes the training of CCS. The algorithm block in Algorithm 1 [1] shows the process of GPI-LS, and from line 3-12, it is very clear that new policies are iteratively added into the policy set $\Pi$. Although in each iteration, if multiple corner weights are selected (line 7), the policies in this iteration can be trained parallelly, while the overall process of GPI-LS is not in parallel.
>
> **(C-2)** Thanks for your suggestion, we will consider revising the notation.
>
> **(C-3)** Thanks for pointing this out and we havel added the constraint on $\omega_i$.
>
> **(C-4) Projection to a compact and convex set.** Proposition 4.4 is related to the proof of convergence of dual problem (2), which is a two-timescale stochastic approximation process. The projection operator is a commonly used technique for constrained optimization to ensure feasibility. The operator follows projections onto convex sets (POCS) or the alternating projection algorithm, which can be implemented very efficiently to ensure both $\theta$ and $\lambda$ are feasible for problem (2).
>
> **(C-5) Updates of min max problem.** Yes, $\pi_r$ is a parameterized policy. The fixed point means a local minima. The iterative steps in Proposition 4.4 describe the updates on both the primal variable ($\pi_r$) and dual variables ($\lambda_r$). It follows the standard primal-dual fixed point referenced as [2] in Appendix. C. The essence of this update rule is to ensure both feasibility with respect to constraints for each other objective in problem (1), and convergence to the fixed point. In numerical optimization, the convergence proof is based on fixed-point iteration, while in the revised paper, we will further clarify that fixed-point found by Proposition 4.4 is where the policy network converged to.
>
> **(C-6) Slater’s condition.** Slater’s condition is a fundamental condition in optimization theory. Slater's condition requires that there exists a feasible policy $\pi$ such that all inequality constraints are strictly satisfied for problem (1), i.e., $$G_i^\pi > d_i, \quad \forall i = 1, \ldots, n, \; i \neq l.$$. This condition ensures that the duality gap between problem (1) and its dual problem (2) is zero, as stated in Theorem 4.5.
>
> **(C-7) Buffer size.** Thank you for pointing this out. In our implementation, the buffer size represents the maximum number of policies that can be stored. If the number of policies exceeds this limit, we store policies based on their crowd distance to match the buffer size. This introduces a trade-off between the diversity of the Pareto set and memory usage: increasing the buffer size allows for a more diverse Pareto set but also results in higher memory costs.
>
> In practice, for many benchmarks, the number of derived Pareto optimal policies is often smaller than the buffer size, meaning that storing them does not require excessive memory resources. Furthermore, as shown in Tables 12 and 13 (please see Table 12 and 13 in ‘Official comment by Authors’ for all reviewers), even with reduced buffer sizes, C-MORL demonstrates highly competitive performance. Compared to PG-MORL, which uses a buffer size of 100 for 2-objective benchmarks and 200 for 3-objective benchmarks, our C-MORL achieves superior results while requiring significantly fewer policies. Another notable observation is that, in some benchmarks, the results remain consistent regardless of the buffer size. This consistency indicates that the number of Pareto optimal policies does not exceed the buffer size in these cases.
>
> **(C-8) Selection of $\beta$.** $\beta G_i^{\pi_r}$ in Eq.3 is related to the corresponding term $d_i$ in Eq.1, which is the threshold of constraint-specific reward functions. Here $G_i^{\pi_r}$ is derived from the latest constrained optimization step, while $\beta$ is a tunable parameter that controls the value of constraint. In Proposition 4.3, we give the sufficient condition of $d_i$, under which the solution of Eq.1 is a Pareto-optimal solution.

---

> > ### Author Response · Authors · 2024-11-23
> >
> > **(C-9) **Selection of $t$.** The core idea of IPO is to augment the objective function with a logarithmic barrier function, enabling the maximization of long-term rewards while satisfying the constraints. In mathematical optimization, the logarithmic barrier function serves as an approximation of the indicator function. Specifically, when the constraints in Eq. (3) are satisfied, the optimization focuses solely on the currently selected objective. However, if the constraints are violated, the penalty becomes $-\infty$, prompting the policy to prioritize maintaining the objective values of corresponding objectives first.
> > A larger value of $t$ offers a closer approximation to the indicator function but also results in higher gradient fluctuations near the boundary of the feasible region, making it more challenging for the policy to satisfy the constraints. Empirically, we follow the recommendation in [3] and set $t=20$. We have mentioned this selection in Appendix E.2.
> >
> > **(C-10) Time complexity.** Proposition 5.1. provides the expected running time for a given $N$. One of the key advantages of C-MORL is its flexibility, allowing a trade-off between computational complexity and performance by selecting an appropriate number of extension policies. Our experimental results empirically demonstrate that even with a limited number of extension policies $N$, C-MORL achieves outstanding performance on benchmarks with up to 9 objectives. As is shown in Table. 2, PG-MORL and GPI-LS failed to solve the 9-objective MORL cases, while C-MORL still solved it in a timely manner.
> >
> > **(C-11) C-MORL-IPO and C-MORL-CPO.** As stated in Appendix E.2, We implement all of our experiments for C-MORL based on C-MORL-IPO. This is because IPO is a first-order method, which is more computationally efficient than C-MORL-CPO. And these two methods’ major differences are on the solving procedure of optimization rather than on the solution quality. In the revised manuscript, we made further explanations on the experiment settings.
> >
> > **(C-12) PG-MORL solves knapsack problem.** We want to note that such a technique is highlighted twice in PG-MORL [4] paper as a key method for conducting prediction-guided optimization for task selection. In Section 3.4, it mentioned Prediction-Guided Optimization for Task Selection: ‘...solve a knapsack problem: given $K \times|\mathcal{P}|$ candidate points in the objective space…’. While in Section A.3 (Supplementary Material), Time Complexity Analysis for Prediction-Guided Optimization: ‘we adopt a greedy algorithm to solve a knapsack problem to select the tasks that can best improve Pareto quality’. And empirically we also found such a method is more time consuming as shown in our Table. 2.
> >
> > **(C-13)** Thank you for pointing this out. In our revised version, a method is also marked if its mean value lies within the confidence interval of the best method. For the HV and EU metrics, our method outperforms competing approaches on all benchmarks. In some benchmarks, the error ranges overlapped with a baseline method, indicating comparable performance. This demonstrates that C-MORL effectively discovers a high-quality Pareto front and achieves high expected utility across diverse user preferences and domains,  particularly in benchmarks with large state and action spaces (MO-Humanoid-2d) and a substantial number of objectives (Building-9d).
> >
> > **(C-14) Experiment setup details.** We ran five times for each experiment for Table 1 and 2. For CAPQL/GPI-LS/Envelope, we use the implementation and default hyperparameters from morl-baselines [5], which is a standard baseline and benchmark toolkit for fair comparison. For PG-MORL/Q-Pensieve, we use the code from the github repo provided in their paper. It is worth emphasizing that compared to PPO, SAC is a much more sample-efficient method because they are off-policy methods, and we believe that is the reason the authors choose them for their sample-efficient MORL approach. For metrics evaluation, we uniformly generate an evaluation preference set in a systematic manner with specified intervals $\Delta=0.01, \Delta=0.1$, and $\Delta=0.5$ for benchmarks with two objectives, three or four objectives, and six or nine objectives, respectively.
> >
> > **(C-15)** For ablation study on Policy selection, each hyperparameter is tested across three runs due to the limitation of computational resources at the time we submit this paper, and results are consistent based on our observations. And we are happy to test across more runs.
> >
> > **(C-16)** It is not clear how to include the ablation study of ‘Pareto Extension procedure and comparing it with solving for random linear weights’. If we remove Pareto extension, and just solve for random linear weights, do you mean we just increase the number of policies for Pareto initialization (with random linear weights)? We provide additional results in Table.3 in our revised manuscript.

---

> > > ### Author Response · Authors · 2024-11-23
> > >
> > > **(C-17) Comparison with IPRP.** In the revised version, we cited this paper and discussed.
> > >
> > > The core idea of IPRO [6] is to bound the search space that may contain value vectors corresponding to Pareto optimal policies and iteratively remove sections from this space. As far as we know, this approach suffers from practical challenges:
> > >
> > > **Large search space.** In IPRO, the iterative updating of the search space is required. However, especially at the beginning, the search space can be excessively large, offering limited guidance for discovering new policies effectively.
> > >
> > > **Obtain Pareto optimal policies.** In IPRO, new policies are derived by solving a standard RL problem for a given linearization vector. This requires training from scratch, which is highly inefficient. In contrast, C-MORL performs constrained policy optimization on the current policies, avoiding the inefficiency of training from the beginning.
> > >
> > > [1] Alegre, Lucas N., et al. "Sample-efficient multi-objective learning via generalized policy improvement prioritization." arXiv preprint arXiv:2301.07784 (2023).
> > >
> > > [2] Vivek S Borkar. Stochastic Approximation:a dynamical systems viewpoint,volume 48. Springer, 2009.
> > >
> > > [3] Liu, Yongshuai, Jiaxin Ding, and Xin Liu. "IPO: Interior-point policy optimization under constraints." Proceedings of the AAAI conference on artificial intelligence. Vol. 34. No. 04. 2020.
> > >
> > > [4] Xu, Jie, et al. "Prediction-guided multi-objective reinforcement learning for continuous robot control." International conference on machine learning. PMLR, 2020.
> > >
> > > [5] Felten, Florian, et al. "A toolkit for reliable benchmarking and research in multi-objective reinforcement learning." Advances in Neural Information Processing Systems 36 (2024).
> > >
> > > [6] Röpke, Willem, et al. "Divide and Conquer: Provably Unveiling the Pareto Front with Multi-Objective Reinforcement Learning." arXiv preprint arXiv:2402.07182 (2024).

---

> > > > ### Comment · Reviewer_dWup · 2024-11-23
> > > >
> > > > I thank the authors for their careful response. Below, I have a few more commentary:
> > > >
> > > > (C-1) Algorithm 1 in [1] is a general overview of the idea of GPI-LS. The practical implementation is shown in Algorithm 2, and this algorithm trains a set of weight vectors in parallel at each iteration (see line 8 and line 22, for instance). Hence, GPI-LS can be parallelized in the same way as C-MORL in the sense that both optimizes some subset of policies at each iteration.
> > > >
> > > > (C-4) and (C-6) Please elaborate on that in the paper for improved clarity.
> > > >
> > > > (C-10) Can you elaborate on why PGMORL and GPI-LS didn't scale with the number of objectives? If they did not finish the training under the time budget, I suggest the authors show the performance that the methods achieved during the limited training. Moreover, computing the hypervolume is extremely slow for larger $d$, and the algorithms in morl-baselines evaluate w.r.t. hypervolume in each iteration. Is it possible that the slow training time is due to the evaluation and not the algorithm per se?
> > > >
> > > > (C-10) Why do you say that C-MORL "solved" the problem? Is the solution it obtained close to optimal? If not, the term "solved" is not adequate.
> > > >
> > > > (C-17) "However, especially at the beginning, the search space can be excessively large, offering limited guidance for discovering new policies effectively."
> > > > In IPRO, the more open space, the better since you can make large improvements, i.e. finding a Pareto optimal point in an open space allows you to exclude a lot of new space.
> > > >
> > > > "In IPRO, new policies are derived by solving a standard RL problem for a given linearization vector."
> > > > This is incorrect. The main idea of IPRO is to avoid linearization vectors. Instead, they apply a Chebyshev scalarisation function which allows you to find points on the concave part by varying a reference point
> > > >
> > > > "This requires training from scratch, which is highly inefficient."
> > > > It’s not required and is not what IPRO does in practice
> > > >
> > > > I suggest the authors be more careful when discussing previous works, to not claim incorrect ideas about competing methods.

---

> ### Author Response · Authors · 2024-11-25
> **Official Comment (New)**
>
> **(C-1)** Thanks for your explanation. In our initial response, we mentioned that ‘ if multiple corner weights are selected (line 7), the policies in this iteration can be trained parallelly’. We believe we are on the same page regarding this point. By ‘not parallel’, we simply mean that the policy training in the next iteration depends on the number of corner weights derived in the last iteration, in this sense it is an iterative process. To avoid confusion, we have modified this expression in our new manuscript.
>
> **(C-4)** and **(C-6)** Thanks for your suggestion. We have revised them in Section 4 and Appendix D.
>
> **(C-10)** PG-MORL requires solving knapsack problems. As mentioned in their supplementary material, the time complexity of PG-MORL is computed by $\mathcal{O}\left(n K|\mathcal{P}| N^{m-2} \log N / p\right)$ when $m>2$, where $m$ is the number of objectives in their paper. From this equation, the time complexity increases exponentially when the number of objectives increases. When running the PG-MORL code, we found that the program got stuck at the prediction-guided optimization for task selection. This is because, with 9 objectives, the computation in this part becomes extremely time-consuming.
>
> We conduct CAPQL and GPI-LS based on morl-baselines [5]. It costs around 69.4 hours to train building-9 for CAPQL. Therefore, we believe that the time spent on evaluation is not the primary factor affecting the actual training time. Through debugging the GPI-LS, we discovered that the program halts at the step where linear_support calculates the next weights (w = linear_support.next_weight()). Due to the limitation of time, we are unable to improve the implementation of GPI-LS for the benchmark with 9 objectives. Therefore, in our current version, we only provide the experiment results for GPI-LS on benchmarks with less than 9 objectives.
>
> **(C-17)** Thanks for your explanation and clarification of the method comparisons.
> However, from equation $s_{\boldsymbol{r}}(\boldsymbol{v})=\min _{j \in\{1, \ldots, d\}} \lambda_j\left(\boldsymbol{v}_j-\boldsymbol{r}_j\right)+\rho\sum\lambda_j\left(\boldsymbol{v}_j-\boldsymbol{r}_j\right)$ , we can see that for the Chebyshev scalarisation, the key step is to still utilize a linearized form to scale $\left(\boldsymbol{v}_j-\boldsymbol{r}_j\right)$. In addition, we would like to clarify that the IPRO used Chebyshev scalarization, but this method can only find solutions on the convex hull of the Pareto front, and all solutions on the non-convex parts of the Pareto front are still missing. From the computation perspective, we echo the reviewer’s claims that “ In IPRO, the more open space, the better since you can make large improvements.” This also draws a connection to our policy initialization stage, where our Assumption 4.2 also guarantees such initial policies are feasible with respect to multiple constraints imposed in Equation (1). Yet we want to emphasize that our policy improvement via Pareto extension is still novel, which is based on the principles of constrained policy optimization, and in our approach we do not take an approximation of Achievement scalarising functions. Following the reviewer’s suggestions, we make a more careful discussion of the IPRO method in the Related Work section: “. (R¨ opke etal.,2024) discovers the Pareto front by bounding the search space that could contain value vectors corresponding to Pareto optimal policies using an approximate Pareto oracle, and iteratively removing sections from this space. Our proposed C-MORL distinguishes from their divide-and-conquer scheme by a novel policy selection procedure along with explicitly solving a principled constrained optimization.”
>
> In addition, we just want to highlight in Table 12 and 13, we can observe that even if the buffer size is significantly decreased, C-MORL still achieves outstanding performance across multiple benchmarks. Another observation is that the results remain consistent in some benchmarks regardless of buffer size, indicating that the number of Pareto optimal policies has not exceeded the buffer size. This further suggests that, in many benchmarks, C-MORL does not require substantial resources to store a large number of policies. These results also show that performances on HV and EU are not sensitive to buffer size for C-MORL. This is because C-MORL fills the gaps on the Pareto front by identifying the sparse regions with crowd distance-based policy selection and the novel design of the constrained policy optimization-based Pareto extension. Although SP deteriorates as the buffer size decreases, it is important to emphasize that SP does not always indicate better performance, for instance, see CAPQL in Fig 4. (b).
>
> We hope this could justify the novelty and contributions of our work.

---

> > ### Author Response · Authors · 2024-11-25
> >
> > Thank you for your feedback and for reviewing our work. We want to kindly check if there are any additional concerns or comments from your side. Please feel free to let us know if we can provide further clarification. Thank you again for your time and efforts.

---

> ### Author Response · Authors · 2024-11-27
>
> Thank you once again for your constructive feedback. We have addressed your concerns in our updated responses and the revised manuscript. Here is a summary of our updates:
>
> **Discussion about baselines:**
>
> **(C-1) GPI-LS:** We have revised the discussion of GPI-LS in our revised manuscript, and also "official comment (new)".
>
> **(C-10) and (C-12) PG-MORL:** We have explained the time complexity of PG-MORL in “official comment (new)“. We have also replaced the verb ‘solve’ with ‘find a satisfiable solution efficiently’.
>
> **(C-17) IPRO:** We have revised the discussion of IPRO in our revised manuscript in Section 2 Related Work, and also “official comment (new)“.
>
> **Explanation of basic RL and optimization concept:**
>
> **(C-4), (C-5), and (C-6):** We have provided a more detailed explanation in our revised manuscript, and also “official comment (new)“.
>
> **Experiment setting:**
>
> **(C-7) Buffer Size:** We have provided additional experiment results in (All-4) Ablation study on buffer size and also “official comment (new)“.
>
> **(C-8) Selection of $\beta$:** We have further explained the selection of $\beta$ in “official comment (C1-8)“.
>
> **(C-9) Selection of $t$:** We have further explained the selection of $t$ in “official comment (C9-16)” and Appendix E.2.
>
> **(C-11) C-MORL-IPO and C-MORL-CPO:** We have highlighted the relevant statement in Appendix E.2 and “official comment (C9-16)“.
>
> Other discussions of the experiment setting are also provided in “official comment (C9-16)“.
>
> **Typos:**
>
> **(C-3) and (C-13):** these have been corrected in the revised manuscript.
>
> Your feedback is greatly appreciated, and we sincerely look forward to further improving this work and hearing from you soon!

---

### Official Review · Reviewer_JTtx · 2024-11-01

**Soundness:** 3
**Presentation:** 3
**Contribution:** 3
**Rating:** 8
**Confidence:** 2

**Summary:**

This paper introduces a novel approach for solving multi-objective reinforcement learning (MORL) problems using a two-stage algorithm called Constrained MORL (C-MORL). The authors proposes to frame the MORL problem as a Constrained MDP. Specifically, in the first stage, C-MORL trains a diverse population of policies across various preference vectors, initializing a set of initial solutions. The second stage then improves these policies via constrained optimization, guaranteeing a targetted expansion of the covered Pareto front.  The authors addressed challenges with existing methods, including limited Pareto front coverage and low training efficiency. The authors show that C-MORL achieves state-of-the-art results on multiple MORL benchmarks in terms of hypervolume and expected utility metrics.

**Strengths:**

**1. Good presentation**

Firstly, the presentation of the paper is well-done, motivations and preliminaries are well set-up from the beginning, and the flow of the paper was coherent. It was easy to read.

**2. Novelty**

The proposed idea is novel in the *MORL* field. It seems to be inspired by approaches from the MOO field, such as Epsilon-constraint methods, but the crossover is done in a creative way that addresses the unique challenges of sequential decision-making setting in MORL. Most contrained MORL problems are more focussed on the aspects of safety constraints, so formalizing the evolution of policies as a constrained optimization problem is novel in MORL, to my best knowledge.

**3. Theoretical rigor**

The paper provides sound propositions and proofs; albeit built upon theoretical foundations already established in CMDPs. One of the more interesting contributions is Proposition 4.3, which provides a practical guideline and sufficient condition for a constrained optimization problem to yield Pareto-optimal solutions. This aspect is a novel adaptation tailored to the requirements of MORL. The condition provided by the proposition is not overly restrictive or computationally intensive to verify. The complexity proof (proposition 5.1) is compelling, and provides strong support for using C-MORL over PGMORL

**4. Extensive experiments**

The authors did a *very* extensive set of experiments across *10* distinct benchmarks, both discrete and continuous. The results are clearly indicative of significant improvements over existing approaches.

**Weaknesses:**

**1. Comparisons against utility-based/preference-conditioned methods can be improved**

The paper mentions that preference-conditioned methods approaches struggle with scalability, I don't fully agree with that. Evolutionary approaches like C-MORL face significant scalability limitations too, for e.g., you can't evolve a population of CNN-based policies (agents) without significant memory overhead. This is worrisome especially if we want to deploy agents to real-world setting with image-based observation spaces. Next, evolutionary approaches seem ideal because you can learn many densely-packed solutions w.r.t. tracing the true optimal Pareto front but they present practical challenges when it comes to policy selection. It is unclear how different users can select the policies they desire for evolutionary approaches whereas utility-based/preference-conditioned methods places these preferences at the forefront. These concerns do not critically impact my overall assessment of this paper, but addressing these points would help clarify the limitations and appropriate use cases of C-MORL for the research community.

**2. No comparisons against the MORL/D algorithm**

The MORL/D algorithm by [1] is very similar to PGMORL and C-MORL, all three of which are evolutionary approaches with populations initialized using preference vectors sampled across the unit simplex. It has state-of-the-art results across many benchmarks, and demonstrates a vast improvement over PGMORL. A significant drawback of PGMORL is perhaps the reliance on prediction models to guide the training, which MORL/D does not depend on. The algorithm is provided in the same code repository as GPI-LS, i.e. [2]. I can imagine the differences between C-MORL and MORL/D in terms of performances being much tighter. Comparisons against this method would be necessary for the results to be more compelling. Specifically, testing against MORL/D with PSA weight adaptation would be very much desired.

[1] Felten, Florian & Talbi, El-Ghazali & Danoy, Grégoire. (2024). Multi-Objective Reinforcement Learning Based on Decomposition: A Taxonomy and Framework. Journal of Artificial Intelligence Research. 79. 679-723. 10.1613/jair.1.15702.

[2] Florian Felten, Lucas N. Alegre, Ann Nowé, Ana L. C. Bazzan, El-Ghazali Talbi, Grégoire Danoy, and Bruno C. da Silva. 2024. A toolkit for reliable benchmarking and research in multi-objective reinforcement learning. In Proceedings of the 37th International Conference on Neural Information Processing Systems (NIPS '23). Curran Associates Inc., Red Hook, NY, USA, Article 1028, 23671–23700.

**Questions:**

**1. How are comparisons between GPI-LS/CAPQL vs PGMORL/C-MORL made?**

As far as I know, CAPQL and GPI-LS are linear scalarization methods and conventionally they are evaluated by sampling preference vectors uniformly across the unit simplex, to form their approximate Pareto front. For evolutionary methods however, they usually keep an archive of policies and their associated vector performances, and then the approximate Pareto front is made during evaluation by filtering the non-Pareto dominated policies. However, this can be concerning during evaluations, especially for metrics like sparsity. If you allow CAPQL/GPI-LS only 100 preference vectors for evaluation, but allow PGMORL/C-MORL and unbounded archive of policies for evaluation, the metrics are naturally going to gravitate in favor of the latter group. Clarification w.r.t. to this aspect needs to be made for me to be more certain about the results of the paper.

**2. How would C-MORL react to concave Pareto fronts?**

In the end of the paper, you mentioned that
> *"...observe that in some benchmarks, the current constrained policy optimization method fails to adequately extend the policies toward certain objective directions"*.

I am curious as to how C-MORL would react to concave pockets on the Pareto fronts? Could the constrained optimization cause the policy to prefer easier paths in convex regions?

**3. Please answer questions in weaknesses**

---

> ### Author Response · Authors · 2024-11-17
>
> Thank you for appreciating the novelty, presentation, theoretical rigor, and experiment of this paper, as well as for your insightful questions.
>
> **(B-1) Comparisons against utility-based/preference-conditioned methods can be improved.**
>
> Thank you for your suggestion.
>
> **Memory resources issue.**
> As you mentioned, typical evolutionary methods suffer from memory resources issue. While C-MORL provides insights to overcome this issue from two perspectives:
> Buffer size $B$ is a tunable parameter that controls the maximum number of policies that can be stored. For real-world tasks with large observation spaces, it allows a trade-off between buffer size and the quality of the Pareto front. From Table 11 and 12, we observe that even with a reduced buffer size, C-MORL maintains highly competitive performance.
> Unlike other evolutionary methods, C-MORL begins the evolutionary process (Pareto extension) to fill the gaps on Pareto front after explicitly optimizing the set of policies that converge to points near the Pareto front. Therefore, for practical applications, we recommend storing a small number of policies and selecting a preferred one for conducting constrained RL-based extensions. This approach enables the derivation of new policies in specific regions of the Pareto front.
>
> **Policy assignment.**
> For ‘policy selection’ (referred to as policy assignment in this paper to avoid confusion) given a specific preference, the user can choose a set-max policy (Definition 3.1). Alternatively, as mentioned earlier, if the Pareto set is not dense enough due to buffer size limitations, constrained RL based extension (C-MORL-IPO in our implementation) can be further performed. This allows extending from a specific policy to obtain diverse policies within a targeted region of the Pareto front.
>
> **(B-2) No comparisons against the MORL/D algorithm.**
>
> Thanks for your suggestion.
>
> **Update:** We have conducted the additional experiments. Please refer to the newly added box for the results and discussion.
>
> **(B-3) How are comparisons between GPI-LS/CAPQL vs PGMORL/C-MORL made?**
>
> We appreciate the reviewer for pointing out the concerns on algorithm evaluations and scalability.
>
> **Buffer size and number of evaluation preference vectors.**
> First, we would like to point out some of our algorithm’s properties and emphasize that buffer size (the number of Pareto optimal policies allowed to store) and number of evaluation preference vectors are different concepts.
> For multi-policy methods, increasing the buffer size allows more Pareto optimal policies to be stored, which generally improves HV and EU but does not guarantee better SP.
> For preference-conditioned policy methods (e.g., linear scalarization methods), increasing the number of evaluation preference vectors tends to improve HV. However, this does not necessarily lead to better EU or SP, as these methods do not ensure that a Pareto optimal policy is assigned to every given preference. For example, in Fig. 4(a), the Pareto front of Q-Pensieve shows that removing the solution on the lower-right corner would lead to better SP (i.e., yield a lower SP value).
> Finally, for multi-policy methods, increasing the number of evaluation preference vectors does not guarantee to improve any of the three metrics.
>
> **Baselines and evaluation of metrics.**
> Among the baseline MORL approaches, CAPQL, Q-Pensieve and GPI-LS are linear scalarization methods. They aim to obtain preference-conditioned policies, and are evaluated by some uniformly sampled preference vectors. PG-MORL/C-MORL are multi-policy methods.
> For multi-policy methods, to evaluate EU, we generate preference vectors with the same interval for all baselines, as is shown in Appendix E.2. To evaluate HV and SP, we directly use the solutions in Pareto set to compute, while we want to emphasize again the number of policies is bounded with a buffer size (100 for 2-objective environments, and 200 for others).
> For linear scalarization methods, we also generate preference vectors with the same interval for all baselines (totally same with multi-policy methods) to evaluate the three metrics, as mentioned in Appendix E.2.
>
> **Additional experiment results.**
> From Table 12 and 13, we can observe that even if the buffer size is significantly decreased, C-MORL still achieves outstanding performance across multiple benchmarks. Another observation is that the results remain consistent in some benchmarks regardless of buffer size, indicating that the number of Pareto optimal policies has not exceeded the buffer size. This further suggests that, in many benchmarks, C-MORL does not require substantial resources to store a large number of policies.
> These results also show that HV and EU are not sensitive to buffer size. Although SP deteriorates as the buffer size decreases, it is important to emphasize that SP does not always indicate better performance, for instance, see CAPQL in Fig 4. (b).

---

> ### Author Response · Authors · 2024-11-17
>
> **(B-4) How would C-MORL react to concave Pareto fronts?**
>
> By mentioning "...observe that in some benchmarks, the current constrained policy optimization method fails to adequately extend the policies toward certain objective directions", the main reason is that in order to ensure fair comparison, we train C-MORL and all baselines with the same steps. Therefore, due to the limited number of steps dispatched for Pareto extension, some areas on Pareto front are not well explored. In the future work, we plan to explore more sample-efficient methods to improve Pareto extension within limited steps for targeted preferences.
>
> Regarding concave pockets on the Pareto fronts, as you pointed out, exploring convex regions are indeed easier for MORL. From the optimization perspective, it is impossible to find a converged and equilibrium solution for concave Pareto front. So that the gradient-based methods could go to few solutions which cannot cover the Pareto fronts well [1].
>
> We also provide a concave benchmark to demonstrate the performance of C-MORL. Deep-sea-treasure-concave is the concave version of Deep-sea-treasure, which is a classic MORL problem in which the agent controls a submarine in a 2D grid world [2]. In this benchmark, the Pareto front is know, so we can compare the approximate Pareto front we discovered with the entire Pareto front. Addition to HV, EU and SP, we use Inverted generational distance (IGD) (the smaller the better), which characterises the convergence rate of an approximate $\mathcal{F}$ towards a reference Pareto front $\mathcal{Z}$ (i.e. the entire Pareto front) [2]:
> $\operatorname{IGD}(\tilde{\mathcal{F}}, \mathcal{Z})=\frac{1}{|\mathcal{Z}|} \sqrt{\sum_{\mathbf{v}^* \in \mathcal{Z}^{\prime}} \min _{\mathbf{v}^\pi \in \tilde{\mathcal{F}}}\left\|\mathbf{v}^*-\mathbf{v}^\pi\right\|^2}$.
>
> **Table 14: Evaluation of HV, EU, SP, IGD for Deep-sea-treasure-concave benchmark.**
>
> |           | HV$(10^{3})$  | EU$(10^{1})$  | SP$(10^{4})$  | IGD$(10^{1})$ |
> |-----------|---------------|---------------|---------------|------------------|
> | GPI-LS    | 3.91$\pm$0.11 | 5.38$\pm$0.00 | 1.32$\pm$0.51 | 1.52$\pm$0.34    |
> | C-MORL    | 4.25$\pm$0.00 | 5.38$\pm$0.00 | 0.05$\pm$0.01 | 0.02$\pm$0.03    |
> | Reference | 4.25$\pm$0.00 | 5.38$\pm$0.00 | 0.04$\pm$0.00 | 0.00$\pm$0.00    |
>
> From Table 14, we observe that compared to GPI-LS, C-MORL achieves higher HV and lower (better) IGD values, indicating a better approximation of the Pareto front. Notably, in two out of five runs, C-MORL successfully discovers the entire Pareto front. However, in the remaining runs, it fails to capture certain neighboring Pareto-optimal solutions. For example, the extire Pareto front is [[1, -1], [2, -3], [3, -5], [5, -7], [8, -8], [16, -9], [24, -13], [50, -14], [74, -17], [124, -19]], while C-MORL obtain an approximation [[1, -1], [5, -7], [8, -8], [16, -9], [24, -13], [50, -14], [74, -17], [124, -19]].
>
>
> [1] Jin, Yaochu, Markus Olhofer, and Bernhard Sendhoff. "Dynamic weighted aggregation for evolutionary multi-objective optimization: Why does it work and how." Proceedings of the genetic and evolutionary computation conference. 2001.
>
> [2] Felten, Florian, et al. "A toolkit for reliable benchmarking and research in multi-objective reinforcement learning." Advances in Neural Information Processing Systems 36 (2024).

---

> > ### Author Response · Authors · 2024-11-24
> >
> > **(B-2) No comparisons against the MORL/D algorithm.**
> >
> > Thanks to your suggestion, we provide the results of MORL/D, implemented using the MORL/D algorithm from the morl-baselines library [2] with PSA weight adaptation. Table 15 demonstrates that C-MORL consistently achieves superior performance across various benchmarks. We would like to highlight once again the key difference between C-MORL and typical evolutionary approaches in MORL. Traditional evolutionary-based MORL methods follow a policy selection and policy improvement framework, where the Pareto front is iteratively improved, and solutions from later iterations typically dominate those from earlier iterations. In contrast, C-MORL employs a different strategy: it first derives a few high-quality Pareto-optimal solutions and then iteratively fills the gaps on the Pareto front.
> >
> > This approach is particularly effective because training for a single preference vector allows the model to converge more easily to a better solution, while the subsequent constrained policy optimization ensures an effective extension of the Pareto front. This combination of localized optimization and systematic expansion is a key factor behind the strong performance of C-MORL.
> >
> > **Table 15: Evaluation of HV, EU, and SP for MORL/D**
> > | Environments                    | Metrics       | MORL/D |
> > |---------------------------------|---------------|--------------------|
> > | MO-Hopper-2d   | HV$(10^{5})$  | 1.11$\pm$0.03      |
> > |                                 | EU$(10^{2})$  | 2.19$\pm$0.04      |
> > |                                 | SP$(10^{2})$  | 2.72$\pm$2.05      |
> > | MO-Hopper-3d   | HV$(10^{7})$  | 1.94$\pm$0.05      |
> > |                                 | EU$(10^{2})$  | 1.68$\pm$0.02      |
> > |                                 | SP$(10^{2})$  | 0.84$\pm$0.17      |
> > | MO-Ant-2d      | HV$(10^{5})$  | 1.03$\pm$0.26      |
> > |                                 | EU$(10^{2})$  | 2.22$\pm$0.49      |
> > |                                 | SP$(10^{3})$  | 2.90$\pm$2.61      |
> > |MO-Ant-3d      | HV$(10^{7})$  | 1.00$\pm$0.17      |
> > |                                 | EU$(10^{2})$  | 1.51$\pm$0.12      |
> > |                                 | SP$(10^{3})$  | 0.85$\pm$0.57      |
> > | MO-Humanoid-2d | HV$(10^{5})$  | 2.81$\pm$0.07      |
> > |                                 | EU$(10^{2})$  | 4.32$\pm$0.06      |
> > |                                 | SP$(10^{4})$  | 0.00$\pm$0.00      |
> > | Building-3d    | HV$(10^{12})$ | 0.87$\pm$0.13      |
> > |                                 | EU$(10^{4})$  | 0.95$\pm$0.00      |
> > |                                 | SP$(10^{4})$  | 73.07$\pm$21.97    |

---

> > > ### Author Response · Authors · 2024-11-25
> > >
> > > Thank you for your feedback and for reviewing our work. We want to kindly check if there are any additional concerns or comments from your side. Please feel free to let us know if we can provide further clarification. Thank you again for your time and efforts.

---

> > > > ### Comment · Reviewer_JTtx · 2024-11-25
> > > >
> > > > Thank you for including the results for MORL/D. The results do look in favor of C-MORL. Please transfer the results into the revised manuscript. Otherwise, the responses from the authors have clarified my queries and retained my confidence in the paper. Thank you. :)

---

> > > > > ### Author Response · Authors · 2024-11-27
> > > > >
> > > > > Thank you for your feedback! We have transferred the updated results for MORL/D into the revised manuscript as you suggested. Please let us know if there’s anything else needed. Thank you once again for your constructive comments and support! :)

---

### Official Review · Reviewer_LYRe · 2024-11-06

**Soundness:** 3
**Presentation:** 1
**Contribution:** 3
**Rating:** 6
**Confidence:** 3

**Summary:**

This paper proposes a two-stage algorithm, Constrained Multi-objective Reinforcement Learning (C-MORL), to efficiently discover the Pareto front in multi-objective reinforcement learning (MORL) problems.  The first stage, Pareto initialization, trains a set of policies in parallel, each with a specific preference vector.  The second stage, Pareto extension, selects policies from the initial set based on their crowd distance and performs constrained optimization to maximize one objective while constraining others.  This approach aims to address limitations in existing MORL methods, such as difficulty covering the complete Pareto front.  The authors demonstrate the effectiveness of C-MORL on various MORL benchmarks, including discrete and continuous control tasks with up to nine objectives, showing superior performance in hypervolume, expected utility, and sparsity compared to baselines.

**Strengths:**

--Originality: The paper introduces a novel perspective by combining constrained policy optimization with MORL, leading to a unique two-stage algorithm for Pareto front discovery.

--Quality: The paper provides a rigorous theoretical analysis of the proposed C-MORL algorithm. The experimental evaluation covers a diverse set of MORL benchmarks with varying state/action spaces and objective dimensions.

--Significance: The proposed C-MORL algorithm provides a more efficient and effective way to discover the Pareto front, especially for complex tasks with high-dimensional objective spaces.

**Weaknesses:**

Clarity: While the core idea is clear, in general it's difficult to follow this paper. The paper could be made more accessible. The algorithm has many stages (Pareto initialization, policy selection, Pareto extension) with different hyperparameters (number of initial policies, number of extension policies, constraint relaxation coefficient) and design choices (C-MORL-CPO vs. C-MORL-IPO, crowd distance vs. random selection). The paper could benefit from a more streamlined presentation of the algorithm and a clearer explanation of the implications of these different choices. Many of these design choices are not studied and the implication of them is not clear. I would suggest the authors move some of the theoretical analysis to the appendix and use the space to discuss these design choices and hyperparameters better.

**Questions:**

How difficult is to determine the specific values for hyperparameters like the number of initial policies, the number of extension policies, and the constraint relaxation coefficient in the experiments? Does this require problem-specific knowledge such as the range of rewards?

---

> ### Author Response · Authors · 2024-11-23
>
> We appreciate your constructive suggestions on improving the manuscript. We have revised the paper accordingly, and in the following, we would like to address the reviewer’s concerns and clarify the paper’s organizations and major contributions.
>
> **(A-1) We revised our manuscript for better presentation.** Thank you for pointing out your concerns about the presentation. We made the description about the C-MORL more concise. In our revised version, we moved some theoretical analysis regarding constrained MORL-based Pareto front extension to the Appendix, and provided further discussion about design choices and hyperparameter study in the experiment section.
>
> **(A-2) Framework of C-MORL.**  The training of C-MORL only includes two stages. In Pareto initialization stage, the goal is to derive a few Pareto optimal policies; then in Pareto extension stage, it performs constrained policy optimization to fill the Pareto front. Specifically, polices are selected to be extended based on their crowd distance.
>
> **(A-3) (Question: how difficult to determine the specific values for hyperparameters?)**
>
> To better understand this problem, we provide additional ablation study.
>
> **Number of initial policies.** We add ablation study for Pareto initialization in the experiment section to understand the influence of number of initial policies. To be specific, we conduct experiments while keeping the total number of training steps fixed at $1.5\times10^6$ steps (including $1\times10^6$ steps for initialization stage) to ensure a fair comparison. We uniformly sample $M=3, 6, 11$ preference vectors, as shown in Table 3. For example, with a sampling interval of $0.2$, the preference vectors are $\boldsymbol{\omega}=[0,1],[0.2, 0.8],[0.4, 0.6],[0.6, 0.4],[0.8, 0.2],[1,0]$. Intuitively, increasing $M$ can enhance diversity among Pareto-optimal solutions, which benefits the subsequent extension phase. However, since the number of total time steps is fixed, increasing $M$ reduces the training steps allocated to each initial policy. The trade-off is evident as the performance for $M=11$ is worse than for $M=6$, indicating that the increasing of $M$ does not always guarantee better performance. To further investigate, we increase the number of training steps to $3\times10^6$ (with $2\times10^6$ steps allocated to initialization stage) while keeping $M=6$. The results demonstrate that proportionally increasing the total training steps and the number of initial policies leads to improved performance, highlighting the importance of balancing training resources with policy diversity.
>
> **Number of extension policies.** In our revised manuscipt, we provide ablation study on number of extension policies in Table 12 on the MO-Ant-3d benchmark, which is one of the continuous MORL tasks with numerous objectives. The number of time steps is proportionally increased with the number of extension policies increases. We can observe that when $N=12$, C-MORL can derive a Pareto front with much higher hypervolume and lower sparsity than that of $N=6$. When $N=18$, the result is similar, indicating that $12$ policies are sufficient to be extended to fill the Pareto front for this task. Fig. 7 further illustrates the Pareto extension process with varying values of the number of extension policies ($N = 6, 12, 18$) on the MO-Ant-3d benchmark. It can be observed that as the number of policies selected to be extended increases, the hypervolume also increases.
>
> **Constraint relaxation coefficient.** Parameter study for return constraint hyperparameter $\beta$ is discussed in the experiment section to examine how constraint values influence the generation of Pareto-optimal solutions. Table 4 presents the results of C-MORL with various $\beta$ on the MO-Ant-2d benchmark. When $\beta = 0.9$, C-MORL presents the best performance on all metrics. Intuitively, a higher $\beta$ ensures that when optimizing a specific objective, the expected returns on other objectives do not decrease significantly, thereby facilitating adaptation to new Pareto-optimal solutions.
> **C-MORL-CPO vs. C-MORL-IPO. ** As stated in Appendix E.2, , we implement all of our experiments for C-MORL based on C-MORL-IPO. This is because IPO is a first-order method, which is more computationally efficient than C-MORL-CPO. And these two methods’ major differences are on the solving procedure of optimization rather than on the solution quality. In the revised manuscript, we made further explanations on the experiment settings.

---

> > ### Author Response · Authors · 2024-11-23
> >
> > **Crowd distance vs. random selection.** Policy selection is a key techinique used in Pareto extension stage, it can efficiently identify the sparse regions on Pareto front to conduct Pareto extension. We compare it with random selection in the experiment section in our revised manuscipt. Table 5 presents the comparison results, where \emph{Random} and \emph{Crowd} refer to random policy selection and crowd distance based policy selection. Each method is tested across three runs with the same seeds to ensure consistent Pareto initialization and uses the same hyperparameters for the Pareto extension. The experimental results show that C-MORL with crowd distance based policy selection outperforms the random selection across all metrics, highlighting the effectiveness of the proposed policy selection method. Intuitively, adjusting the policies in sparse areas facilitates better filling of the gaps in the Pareto front.
> >
> > **(A-4) (Question: Does this require problem-specific knowledge such as the range of rewards?)**
> >
> > Based on our experiment results, C-MORL demonstrates robustness across various benchmarks. Unlike some methods, it does not require problem-specific knowledge, such as the range of rewards. For practical applications, the hyperparameters can be adjusted to accommodate changes in the number of objectives. For example, as the number of objectives increases, we recommend increasing the training steps, the number of initialization and extension policies, as well as the policy buffer size.
> >
> > Regarding the range of rewards, slight imbalances among objectives may lead to Pareto solutions derived from Pareto initialization being relatively concentrated in specific regions of the Pareto front. In such cases, our crowd distance-based policy selection ensures that policies in sparse areas of the Pareto front have a higher priority of being chosen and extended, helping to fill gaps. However, in extreme cases where the reward ranges are highly imbalanced, we suggest assigning higher weights to objectives with lower reward ranges when selecting preference vectors during the initialization stage. This adjustment improves practical usability by promoting better coverage of the Pareto front.

---

> > > ### Author Response · Authors · 2024-11-25
> > >
> > > Thank you for your feedback and for reviewing our work. We want to kindly check if there are any additional concerns or comments from your side. Please feel free to let us know if we can provide further clarifications. Thank you again for your time and efforts.

---

> > > > ### Author Response · Authors · 2024-11-27
> > > >
> > > > Thank you again for your constructive feedback. We have addressed the concerns you raised about clarity and the choice of hyperparameters, and revised the manuscript accordingly. Let us summarize our updates as follows:
> > > >
> > > > **Streamlined Presentation:** We have moved the theoretical analysis of constrained policy optimization to Appendix F to improve the flow of the main text. We also refined the descriptions of algorithm steps to make C-MORL easier to follow.
> > > >
> > > > **Hyperparameter Discussion:** We have expanded the discussion on hyperparameter choices in the experiment section and in our previous response. This includes details about the number of initial policies, extension policies, the constraint relaxation coefficient, crowd distance versus random selection, and the choice between C-MORL-CPO and C-MORL-IPO. Hope this gives a more comprehensive view of C-MORL, which we observe is quite robust to reward range and constraint relaxation coefficients.
> > > >
> > > > **Problem-Specific Knowledge:** Your question about the requirement for problem-specific knowledge, such as the range of rewards, has also been addressed in our earlier reply.
> > > >
> > > > Your feedback is highly valuable to us, and we sincerely hope to hear from you soon :)

---

> > > > > ### Comment · Reviewer_LYRe · 2024-11-28
> > > > >
> > > > > Thank you for the detailed responses and the updates. My concerns are addressed so I adjust my score accordingly.

---

> > > > > > ### Author Response · Authors · 2024-11-28
> > > > > >
> > > > > > Thank you for recognizing the improvements in our paper. We greatly appreciate your constructive feedback and support.

---

### Author Response · Authors · 2024-11-17

We sincerely thank all the reviewers for their valuable feedback and suggestions. Here we would like to respond to reviewers' shared concerns and comments:

**(All-1) Contributions and novelty.**

**Framework.** The key innovation of our algorithm lies in its ability to efficiently construct the full Pareto front by constrained policy optimization-based Pareto extension, which starts from only a small number of Pareto optimal solutions by training a limited set of policies. This allows us to utilize rich tools and theories from constrained optimization to MORL problems.

**Training efficiency.** From only a small number of Pareto optimal policies, C-MORL employs a constrained policy optimization-based Pareto extension to systematically derive new policies, completing the Pareto front without the need to train additional policies from scratch. This approach significantly reduces computational overhead while ensuring high-quality solutions across the entire Pareto front.

**Ability to extend to benchmarks with large objective spaces.** Existing methods primarily focus on tasks with continuous observation spaces and up to 5 objectives, or discrete observation spaces with up to 6 objectives. In contrast, C-MORL can effectively handle benchmarks with larger objective spaces (up to 9 objectives in our experiments) by identifying gaps on the Pareto front and generating new policies in the desired regions. This is achieved by optimizing a policy toward a specific objective direction while constraining the performance of other objectives, enabling precise and efficient expansion of the Pareto front.

**(All-2) Simulation settings and comparisons.** In our revised version, we further clarify the simulation settings. As stated in Appendix E.2 , we implement all of our experiments for C-MORL based on C-MORL-IPO. In the Pareto initialization stage, the preference vectors used to guide the training of initial policies are uniformly sampled. In the Pareto extension stage, the number of policies is bounded with a buffer size (100 for 2-objective environments, and 200 for others). Regarding evaluation, a method is also marked if its mean value lies within the confidence interval of the best method. Based on the updated Table. 1 and Table. 2, we can clearly see proposed C-MORL achieves consistent performances and almost always the best performances across all metrics and all testing environments.

**(All-3) Paper organizations.** To improve the presentation of this paper, we moved some theoretical analysis regarding constrained MORL-based Pareto front extension to the Appendix. In the main text, we follow reviewers’ suggestions and i). Make the description of our approach easy to follow; and ii). We also provide further discussion about design choices and hyperparameter study in the Experiment section, including parameter study of C-MORL for Pareto Initialization, parameter study for return constraint hyperparameter $\beta$, ablation study on policy selection. Further ablation studies on number of extension policies and buffer size are provided in Appendix G.

---

> ### Author Response · Authors · 2024-11-17
>
> **(All-4) Ablation study on buffer size.**
> In order to better understand how Buffer size can influence the performance of C-MORL, we provide the comparison of the best baseline v.s. C-MORL with various buffer sizes.
>
> **Table 12: Evaluation of HV, EU, and SP for discrete MORL tasks under varying buffer sizes, alongside a comparison with the best-performing baseline.**
> | Environments    | Metrics       | Best Baseline                    | B=20                   | B=50                   | B=100                  | B=200                  |
> |-----------------|---------------|----------------------------------|------------------------|------------------------|------------------------|------------------------|
> | Minecart        | HV$(10^{2})$  | **6.05$\pm$0.37(GPI-LS)**        | **6.22$\pm$0.70** | **6.57$\pm$0.92** | **6.63$\pm$0.89** | **6.77$\pm$0.88** |
> |                 | EU$(10^{-1})$ | **2.29$\pm$0.32(GPI-LS)**   | 1.88$\pm$0.59          | **2.05$\pm$0.67** | **2.09$\pm$0.65** | **2.12$\pm$0.66** |
> |                 | SP$(10^{-1})$ | 0.10$\pm$0.00(GPI-LS)            | 0.64$\pm$0.17          | 0.19$\pm$0.02          | 0.09$\pm$0.03          | **0.05$\pm$0.02** |
> | MO-Lunar-Lander | HV$(10^{9})$  | 1.06$\pm$0.16(GPI-LS)            | 1.01$\pm$0.07          | **1.08$\pm$0.04** | **1.12$\pm$0.03** | **1.12$\pm$0.03** |
> |                 | EU$(10^{1})$  | 1.81$\pm$0.34(GPI-LS)            | 1.84$\pm$0.31          | **2.21$\pm$0.23** | **2.35$\pm$0.18** | **2.35$\pm$0.18** |
> |                 | SP$(10^{3})$  | **0.13$\pm$0.01(GPI-LS)**   | 1.14$\pm$2.14          | 1.65$\pm$1.83          | 1.04$\pm$1.24          | 1.04$\pm$1.24          |
> | Fruit-Tree      | HV$(10^{4})$  | **3.66$\pm$0.23(Envelope)** | 2.34$\pm$0.29          | 2.86$\pm$0.19          | 3.17$\pm$0.20          | **3.67$\pm$0.14** |
> |                 | EU            | 6.15±0.00(Envelope/GPI-LS)       | 6.14$\pm$0.13          | 6.38$\pm$0.10          | 6.46$\pm$0.08          | **6.53$\pm$0.03** |
> |                 | SP            | 0.53±0.02(GPI-LS)                | 2.62$\pm$0.49          | 1.65$\pm$1.83          | 0.81$\pm$0.12          | **0.04±0.00**     |

---

> > ### Author Response · Authors · 2024-11-23
> >
> > **Table 13: Evaluation of HV, EU, and SP for continuous MORL tasks under varying buffer sizes, alongside a comparison with the best-performing baseline.**
> > | Environments   | Metrics       | Best Baseline                 | B=20              | B=50              | B=100             | B=200             |
> > |----------------|---------------|-------------------------------|-------------------|-------------------|-------------------|-------------------|
> > | MO-Hopper-2d   | HV$(10^{5})$  | 1.26$\pm$0.01(Q-Pensieve)     | **1.39$\pm$0.01** | **1.39$\pm$0.01** | **1.39$\pm$0.01** | -                 |
> > |                | EU$(10^{2})$  | 2.34$\pm$0.10(PG-MORL)        | **2.55$\pm$0.01** | **2.56$\pm$0.02** | **2.56$\pm$0.02** | -                 |
> > |                | SP$(10^{2})$  | **0.46$\pm$0.10(CAPQL)**      | 2.68$\pm$1.66     | **0.57$\pm$0.29** | **0.33$\pm$0.28** | -                 |
> > | MO-Hopper-3d   | HV$(10^{7})$  | 1.70$\pm$0.29(GPI-LS)         | 2.03$\pm$0.15     | 2.20$\pm$0.04     | 2.26$\pm$0.02     | **2.29$\pm$0.01** |
> > |                | EU$(10^{2})$  | 1.62$\pm$0.10(GPI-LS)         | 1.72$\pm$0.08     | 1.78$\pm$0.02     | **1.80$\pm$0.01** | **1.80$\pm$0.01** |
> > |                | SP$(10^{2})$  | 0.74$\pm$1.22(GPI-LS)         | 7.61$\pm$2.24     | 2.74$\pm$1.91     | 0.86$\pm$0.27     | **0.28$\pm$0.09** |
> > | MO-Ant-2d      | HV$(10^{5})$  | **3.10$\pm$0.25(GPI-LS)**     | **3.08$\pm$0.21** | **3.13$\pm$0.20** | **3.13$\pm$0.20** | -                 |
> > |                | EU$(10^{2})$  | **4.28$\pm$0.19(GPI-LS)**     | **4.27$\pm$0.19** | **4.29$\pm$0.19** | **4.29$\pm$0.19** | -                 |
> > |                | SP$(10^{3})$  | **0.18$\pm$0.07(CAPQL)**      | 3.66$\pm$1.29     | 1.67$\pm$0.85     | 1.67$\pm$0.85     | -                 |
> > | MO-Ant-3d      | HV$(10^{7})$  | 3.82$\pm$0.43(Q-Pensieve)     | 3.52$\pm$0.16     | 3.83$\pm$0.17     | **4.00$\pm$0.12** | **4.09$\pm$0.13** |
> > |                | EU$(10^{2})$  | 2.41$\pm$0.20(GPI-LS)         | 2.48$\pm$0.09     | **2.55$\pm$0.08** | **2.57$\pm$0.07** | **2.57$\pm$0.06** |
> > |                | SP$(10^{3})$  | **0.02$\pm$0.01(PG-MORL)**    | 1.56$\pm$0.67     | 0.30$\pm$0.19     | 0.08$\pm$0.05     | **0.03$\pm$0.01** |
> > | MO-Humanoid-2d | HV$(10^{5})$  | 3.30$\pm$0.05(CAPQL)          | **3.43$\pm$0.06** | **3.43$\pm$0.06** | **3.43$\pm$0.06** | -                 |
> > |                | EU$(10^{2})$  | **4.75$\pm$0.04(CAPQL)** | **4.78$\pm$0.05** | **4.78$\pm$0.05** | **4.78$\pm$0.05** | -                 |
> > |                | SP$(10^{4})$  | **0$\pm$0.00(CAPQL/GPI-LS)**  | 0.18$\pm$0.27     | 0.18$\pm$0.27     | 0.18$\pm$0.27     | -                 |
> > | Building-3d    | HV$(10^{12})$ | 1.00$\pm$0.02(Q-Pensieve)     | 1.14$\pm$0.00     | 1.14$\pm$0.00     | **1.15$\pm$0.00** | **1.15$\pm$0.00** |
> > |                | EU$(10^{4})$  | 0.96$\pm$0.00(Q-Pensieve)     | **1.02$\pm$0.00** | **1.02$\pm$0.00** | **1.02$\pm$0.00** | **1.02$\pm$0.00** |
> > |                | SP$(10^{4})$  | **0.37$\pm$0.22(PG-MORL)**    | 1.04$\pm$0.16     | 1.98$\pm$0.38     | 0.69$\pm$0.62     | 0.69$\pm$0.62     |
> > | Building-9d    | HV$(10^{31})$ | 7.28$\pm$0.57(Q-Pensieve)     | 7.64$\pm$0.17     | **7.93$\pm$0.07** | **7.93$\pm$0.07** | **7.93$\pm$0.07** |
> > |                | EU$(10^{3})$  | 3.46$\pm$0.03(Q-Pensieve)     | 3.50$\pm$0.00     | **3.52$\pm$0.00** | **3.52$\pm$0.00** | **3.52$\pm$0.00** |
> > |                | SP$(10^{4})$  | **0.10$\pm$0.04(Q-Pensieve)** | 1.16$\pm$0.19     | 0.28$\pm$0.03     | 0.28$\pm$0.04     | 0.28$\pm$0.04     |

---

### Meta-Review · Area_Chair_hkg4 · 2024-12-21

**Metareview:**

This paper introduces C-MORL, a method for approximating the Pareto front in multi-objective reinforcement learning through constrained optimization.
The reviewers appreciated the relevance of the problem and the novelty of the proposed approach, as well as the extensive experimental evaluation across challenging benchmarks. The authors’ rebuttal addressed many concerns, including clarifying methodological details and providing additional results, which strengthened confidence in the paper’s contributions. While some areas, such as computational efficiency and clarity, could benefit from further refinement, the work represents a meaningful advancement in multi-objective reinforcement learning.
Overall, the paper makes a valuable contribution and merits acceptance.

**Additional Comments On Reviewer Discussion:**

During the rebuttal period, the reviewers raised concerns regarding methodological clarity, computational efficiency, experimental robustness, and novelty. The authors addressed these by providing detailed clarifications, integrating additional results (e.g., MORL/D), and revising explanations to improve the presentation and align their work with prior methods.

Most reviewers acknowledged the improvements and adjusted their scores positively, citing stronger experimental evidence and clearer methodology. However, some concerns about scalability, statistical rigor, and resource efficiency persisted, particularly from the third reviewer, who maintained their score despite appreciating the authors' efforts.

Considering the overall contribution, relevance, and addressed concerns, the paper presents valuable advancements in MORL, and the updates justify its acceptance.

---

### Decision · Program_Chairs · 2025-01-22

Accept (Poster)